# NOISE-AWARE GENERALIZATION: ROBUSTNESS TO IN-DOMAIN NOISE AND OUT-OF-DOMAIN GENERALIZATION

## ABSTRACT

Training on real-world data is challenging due to its complex nature, where data is often noisy and may require understanding diverse domains. Methods focused on Learning with Noisy Labels (LNL) may help with noise, but they often assume no domain shifts. In contrast, approaches for Domain Generalization (DG) could help with domain shifts, but these methods either consider label noise but prioritize out-of-domain (OOD) gains at the cost of in-domain (ID) performance, or they try to balance ID and OOD performance, but do not consider label noise at all. Thus, no work explores the combined challenge of balancing ID and OOD performance in the presence of label noise, limiting their impact. We refer to this challenging task as Noise-Aware Generalization, and this work provides the first exploration of its unique properties. We find that combining the settings explored in LNL and DG poses new challenges not present in either task alone, and thus, requires direct study. Our findings are based on a study comprised of three real-world datasets and one synthesized noise dataset, where we benchmark a dozen unique methods along with many combinations that are sampled from both the LNL and DG literature. We find that the best method for each setting varies, with older DG and LNL methods often beating the SOTA. A significant challenge we identified stems from unbalanced noise sources and domain-specific sensitivities, which makes using traditional LNL sample selection strategies that often perform well on LNL benchmarks a challenge. While we show this can be mitigated when domain labels are available, we find that LNL and DG regularization methods often perform better.

## 1 INTRODUCTION

As deep learning models grow in complexity, the need for extensive training datasets has increased. However, real-world data collection often introduces noise and aggregates samples from multiple sources, creating challenges for training. To effectively address these issues, it is essential to consider three critical perspectives: in-domain performance, out-of-domain performance, and robustness to label noise, as illustrated in Fig. 1-(a).

*Learning with Noisy Labels (LNL)* addresses the intersection of in-domain performance and noise robustness, aiming to mitigate the impact of incorrect labels in real-world datasets (Natarajan et al., 2013; Arpit et al., 2017; Song et al., 2022; Xia et al., 2021; 2023; Wei et al., 2022; Liu et al., 2021; Song et al., 2024; Cordeiro et al., 2023; Shen & Sanghavi, 2019). However, these methods often assume a single data distribution, having issues with distinct feature distributions when noisy labels coincide with domain shifts, as shown in Fig. 1-(b). *Domain Generalization (DG)* aims to train models that generalize to unseen target domains after learning from multiple source domains (Cha et al., 2022; 2021; Wang et al., 2023; Bui et al., 2021; Arjovsky et al., 2019; Kamath et al., 2021; Chen et al., 2022; 2024a; Rame et al., 2022; Lin et al., 2022; Zhang et al., 2024). While many DG methods focus primarily on out-of-domain performance, a subset also evaluates both source and target domains—termed as *Domain-Aware Optimization* methods (Wortsman et al., 2022; Zhang et al., 2024). However, this group often overlooks the impact of noise and tends to overfit when faced with noisy labels (Qiao & Low, 2024). Additionally, some DG methods show implicit *OOD-robustness* under noise (Rame et al., 2022; Sagawa et al., 2019; Krueger et al., 2021; Qiao & Low, 2024;

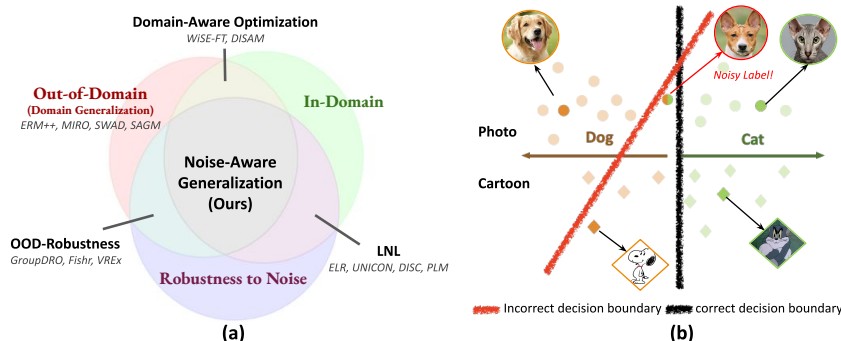

Figure 1: **Comparison to prior work**. (a) The relationship between our task and related works, illustrated by three overlapping circles representing In-Domain Performance, Out-of-Domain Performance (Teterwak et al., 2023; Cha et al., 2022; 2021; Wang et al., 2023), and Robustness to Noise. LNL (Liu et al., 2020; Li et al., 2023; Karim et al., 2022; Zhao et al., 2024), Domain-Aware Optimization (Zhang et al., 2024; Wortsman et al., 2022), and OOD-Robustness (Sagawa et al., 2019; Rame et al., 2022; Krueger et al., 2021) correspond to the intersections between areas *(corresponding methods are listed below)*, with our work at the center, addressing all three aspects. (b) The challenges of Noise-Aware Generalization: Noisy label samples and those from varying (minority) distributions can mislead the model, resulting in inaccurate decision boundaries.

Humblot-Renaux et al., 2024), but often place more emphasis on out-of-domain performance while neglecting the in-domain performance in noisy environments.

By examining related work, as visualized in Fig. 1-(a), we observe that previous research addresses only portions of this problem space. Notably, the intersection where all three aspects—in-domain performance, out-of-domain generalization, and noise robustness—overlap is missing.

To bridge this gap, we introduce **Noise-Aware Generalization**, a novel task designed to capture the complex challenges of training on noisy, multi-domain datasets. In practice, training data is often collected under the assumption that the test data will originate from a similar distribution, making in-domain performance crucial. Meanwhile, real-world applications frequently require models to generalize across diverse domains, highlighting the importance of out-of-domain generalization as well. Additionally, handling label noise is unavoidable, necessitating a focus on robustness to noise. Noise-Aware Generalization emphasizes the intersection of these three critical considerations.

Surprisingly, even the combinations of state-of-the-art LNL and DG methods do not perform well in this setting, indicating that challenges arise when integrating these approaches. We expand our analysis by exploring the effects of multi-distribution data on LNL methods, the sensitivity to noise across different domains, and the balance between domain distribution and label cleanliness. Our study also provides insights into how LNL regularizers can complement DG methods and highlights the potential of leveraging domain labels to enhance sample selection in LNL tasks.

Our contributions are summarized below:
- We propose a new task, Noise-Aware Generalization, which contains both noisy labels with domain shifts and evaluates both on in-domain and out-of-domain performance. We find that combining the best performing LNL+DG from prior work does not generalize well to our setting, suggesting that they have overfit to their respective task assumptions.
- We present a unified framework that integrates DG with LNL methods. Additionally, we provide a rough noise estimation for three real-world datasets with multi-domain data from diverse fields: web/user (Fang et al., 2013), e-commerce (Xiao et al., 2015), and biological images (Chen et al., 2024b). This framework and noise estimation can support future studies on noise robustness and the intersection of DG methods.
- We perform a critical analysis of twenty older and state-of-the-art (SOTA) methods in DG and LNL, along with their combinations. Our experimental settings on Noise-Aware Generalization provide valuable insights for future research in this area.

## 2 NOISE-AWARE GENERALIZATION STUDY

In this section, we begin by formally defining the Noise-Aware Generalization task and presenting a unified framework that integrates both LNL and DG perspectives. We then analyze real-world datasets to demonstrate the existence of Noise-Aware Generalization in practical training scenarios. This section forms the foundational components necessary for conducting comprehensive experiments and analysis in the subsequent sections.

### 2.1 NOISE-AWARE GENERALIZATION FRAMEWORK

Consider a multi-domain dataset $\mathcal{D}$ with $m$ source domains: $\mathcal{D} = \{\mathcal{D}_1, \mathcal{D}_2, \ldots, \mathcal{D}_m\}$, where each $\mathcal{D}_i = \{(x_{i,j}, y_{i,j})\}_{j=1}^{n_i}$ represents samples from domain $i$ with $x_{i,j}$ as the input and $y_{i,j}$ as the label, potentially noisy. During the test, an unseen target domain $\mathcal{D}_{target}$ will be used for OOD-evaluation. The goal is to learn a model $f_\theta(x)$ parameterized by $\theta$ that performs well across all source domains $\{\mathcal{D}_i\}_{i=1}^m$ and generalized to $\mathcal{D}_{target}$, despite the presence of label noise.

**LNL objectives.** The typical loss function for LNL seeks to minimize the impact of label noise, with methods broadly categorized into *non-separating* and *separating*. *Non-separating* methods, such as learning noise transitions (Scott, 2015; Liu & Tao, 2015; Menon et al., 2015; Patrini et al., 2017; Li et al., 2021; Zhang et al., 2021; Kye et al., 2022; Cheng et al., 2022; Liu et al., 2023; Li et al., 2022b; Vapnik et al., 2013; Yong et al., 2022; Zhao et al., 2024), adjust the label with noise transition matrices (Xia et al., 2019; Yao et al., 2020; Yang et al., 2022). *Separating* methods split the training set into subgroups and employ semi-supervised learning (SSL) techniques (Hu et al., 2021; Torkzadehmahani et al., 2022; Nguyen et al., 2019; Tanaka et al., 2018; Li et al., 2022a; Feng et al., 2021). Detecting clean samples include *loss-based* methods that assume samples with large losses are noisy (Jiang et al., 2018; Li et al., 2020; Arazo et al., 2019), *similarity-based* methods identify clean-sample clusters within each class (Mirzasoleiman et al., 2020; Kim et al., 2021). and *data augmentation* (Li et al., 2023; Karim et al., 2022) methods that select clean samples with consistent predictions across different augmentation strengths. After splitting the data into clean and noisy, some methods remove noisy samples from training (Xia et al., 2021; 2023; Wei et al., 2022; Liu et al., 2021; Song et al., 2024; Cordeiro et al., 2023; Shen & Sanghavi, 2019), while others apply SSL (Sohn et al., 2020; Tarvainen & Valpola, 2017; Li et al., 2020; Karim et al., 2022; Li et al., 2023).

More formally, for domain $i$ the weighted empirical risk with noisy labels can be written as:

$$\mathcal{L}_{LNL}^{(i)} = \frac{1}{|\mathcal{D}_i|} \sum_{(x_{i,j}, y_{i,j}) \in \mathcal{D}_i} \omega(y_{i,j}) l(f_\theta(x_{i,j}), \tau(y_{i,j})). \tag{1}$$

This single equation highlights the key aspects across LNL methods. $l(\cdot, \cdot)$ is a loss function such as cross-entropy, and $\omega(y_{i,j})$ is a weight that adjusts the impact of potentially noisy labels, often determined via clean label detection techniques. For example, for *non-separating* methods like ELR (Liu et al., 2020) and PLM (Zhao et al., 2024), $\omega(y_{i,j}) = 1$ for all the samples. While for *separating* methods, such as UNICON (Karim et al., 2022) and DISC (Li et al., 2023), $\omega(y_{i,j})$ varies for clean and noisy subgroups. $\tau(\cdot)$ denotes a label transformation, such as a corrected version of the original label. For example, PLM use the estimated noise transition matrix to transform the noisy labels, UNICON and DISC apply mixup on the noisy subset, where $\tau(y_{i,j})$ is the mixup label.

**DG objectives.** The goal of $\mathcal{L}_{DG}$ is to capture domain-level variations and learn domain-invariant representations, ensuring that the model isn't overly biased toward any single domain during training (Gulrajani & Lopez-Paz, 2021; Li et al., 2017a;b; 2019; 2018a;b; Muandet et al., 2013). By examining differences across domains, it attempts to generalize better to unseen data.

$$\mathcal{L}_{DG} = \sum_{i=1}^m \left( \sum_{j \neq i} \text{Var}(g_j(\theta)) \right). \tag{2}$$

where $g_j(\theta)$ represents domain $j$'s contribution from the parameterized model $f_\theta(\cdot)$. The objective function aims to minimize domain-wise variations by evaluating how the representations differ across domains, thereby learning features that are consistent and robust across different domains. For example, MIRO (Cha et al., 2022) maximizes the mutual information between representations from an oracle model and a trained model, which ensures that the learned representations are consistent

Figure 2: **Real-world datasets with in-domain noise and multi-domain distribution**. **VLCS** (web/user data) (Fang et al., 2013), **Clothing1M** (e-commerce) (Xiao et al., 2015), and **CHAMMI-CP** (biomedical images) (Chen et al., 2024b). VLCS and Clothing1M face label noise from poor annotations and domain shifts from varying data sources, while CHAMMI-CP deals with ambiguous features and varying experimental environments.

across domains, effectively reducing $g_j(\theta)$'s domain-specific variations and thereby achieving better generalization to unseen domains.

**Regularization terms.** *Non-separating* LNL methods often incorporate regularization to prevent the model from memorizing noisy labels, guiding it toward more reliable target probabilities (Liu et al., 2020; 2022a). The regularization term operates on the predicted logits, and a unified form of LNL regularization can be expressed as: $\mathcal{R}_{LNL}^{(i)} = \sum_{j=1}^{n} \phi(p_{i,j}, \tau(y_{i,j}))$, where $p_{i,j}$ is the predicted probability logits for the $j$−th sample. $\phi(\cdot, \cdot)$ is a function to enforce regularization, *e.g.*, $\phi(p_{i,j}, \tau(y_{i,j}) = log(1- <p_{i,j}, \tau(y_{i,j})>)$ in ELR (Liu et al., 2020).

In Domain Generalization (DG), regularization serves as a key component to enhance robustness (Foret et al., 2020), aiming to minimize the worst-case loss in a neighborhood around the model parameters (Cha et al., 2021; Wang et al., 2023; Zhang et al., 2024). This regularization is formulated as: $\mathcal{R}_{DG} = \max_{\|\epsilon\| \leq \rho} L(\theta + \epsilon)$, where $\rho$ controls the perturbation radius.

**Final objective.** The final objective function for Noise-Aware Generalization is:

$$\mathcal{L}_{NG} = \alpha \frac{1}{m} \sum_{i=1}^{m} \mathcal{L}_{LNL}^{(i)} + \beta \mathcal{L}_{DG} + \lambda \mathcal{R}_{LNL} + +\gamma \mathcal{R}_{DG}. \tag{3}$$

where $\alpha$, $\beta$, $\lambda$, and $\gamma$ are hyperparameters that balance the contributions from the LNL loss, DG loss, LNL regularization, and SAM regularization respectively. Our Noise-Aware Generalization integration methods follow the unified framework and detailed algorithms for the methods used in our experiments are provided in Appendix C.2.

## 2.2 NOISE-AWARE GENERALIZATION CHALLENGE IN REAL-WORLD DATASETS

**VLCS** (Fang et al., 2013) is a well-known benchmark used for domain generalization. It consists of images drawn from four distinct datasets: VOC2007 (V) (Everingham et al., 2010), LabelMe (L) (Russell et al., 2008), Caltech101 (C) (Fei-Fei et al., 2004), and SUN09 (S) (Choi et al., 2010). Each dataset represents a different domain with its unique distribution. The primary challenge with VLCS lies in its inherent domain shifts. It also involves the presence of noisy labels, which is overlooked by the prior work. A thorough manual inspection reveals an unbalanced noise distribution across domains. Caltech101 is the cleanest and easiest domain, featuring clear backgrounds and salient objects. However, LabelMe exhibits substantial noise, with over 80% of the "person" images being incorrectly labeled, often depicting cars or street scenes. Similar noise issues are observed in VOC2007 and SUN09, where numerous "car" images are mislabeled as persons, and a majority of "chair" images contain people. Further examples can be seen in Fig. 2, with additional details provided in the Appendix B.

**Clothing1M.** (Xiao et al., 2015) is a benchmark for learning noisy labels. It contains approximately 1 million images of clothing items and 14 clothing categories, where the noise is estimated to affect around 40% of the labels. However, what's overlooked in this dataset, is the domain shift within the training samples, the images in Clothing1M are collected from three distinct online shopping websites, which can be treated as three different data sources. As shown in Fig. 2, the domain shift does exist in the data.

**CHAMMI-CP** (Chen et al., 2024b) is from a collection of approximately 8 million single-cell images, which utilized the Cell Painting assay (Bray et al., 2016),an advanced imaging technique that stains eight cellular compartments using six fluorescent markers, which are then captured in five imaging channels. This dataset plays a crucial role in quantifying cellular responses to various treatments or perturbations, a fundamental process in drug discovery research. The challenges in this dataset involve both noisy labels related to control images and domain shifts under different technical variations in the experiment settings. For control cells, also referred to as the "do-nothing" group, there can be confusion with weak-treatment cells. When the treatment effect is minimal, weak-treatment cells may visually resemble control cells (Bray et al., 2016). In such cases, despite being labeled as weak treatment, their visual features align more closely with control cells. Thus, for treatment classification tasks, these cells should use control as the correct label. Regarding the domain shift observed in these cell images, the cells undergo treatment in various environments (plates), leading to technical variations that introduce domain-specific features.

## 3 EXPERIMENTS

We conduct two types of experiments. First, we evaluate ID and OOD performance on real-world datasets. ID performance is tested on datasets from the training domains. For OOD performance, we follow the "leave-one-out" protocol, leaving one domain out as the test domain and training with the remaining domains. The results reported are the average performance across all test domains. The second type of experiment focused on analyzing the challenges of combining LNL and DG tasks. For this, we include DomainNet (Peng et al., 2019) with synthesized noise to facilitate analysis.

### 3.1 EVALUATION METRICS AND DATASETS

Since the goal of Noise-Aware Generalization is to achieve high accuracy on both ID and OOD data, we report classification accuracy on two test sets for each trained model: an ID-test set with the same distribution as the training set and an OOD-test set from a different domain.

**Datasets.** We use three real-world datasets (shown in Fig. 2) and one synthetic noise dataset. These real-world datasets contain both noisy labels and distribution shifts. For Clothing1M (Xiao et al., 2015), domain labels aren't available, so we can't split it for OOD testing. Instead, we introduce Fashion-MNIST (Xiao et al., 2017) as an OOD test set to evaluate domain generalization. Fashion-MNIST contains 70,000 grayscale images of 10 fashion item categories, each 28x28 pixels, similar to MNIST. We refer to this combination as **Noise-Aware Generalization -Fashion**, using 7 classes from Clothing1M and 5 classes from Fashion-MNIST, all shared between the two datasets.

DomainNet-SN is an additional synthetic noise dataset to complement our real-world datasets. DomainNet (Peng et al., 2019) features over six million images across 6 domains (real photos, sketches, paintings, clipart, infographics, and quickdraw) spanning 345 classes. It provides a diverse range of visual data, enriching our analysis by examining how noise interacts with domain shifts. DomainNet-SN incorporates asymmetric noise, where noisy label pairs are derived from the training confusion matrix. For each class, the target class with the second-highest prediction probability is chosen as the noisy label source. Details about the synthetic noise are provided in appendix A.

### 3.2 RESULTS ON REAL-WORLD DATASETS

Tab. 1 presents the performance of various methods on three different datasets: VLCS (Fang et al., 2013), Noise-Aware Generalization-Fashion (Xiao et al., 2015; 2017), and CHAMMI-CP (Chen et al., 2024b). For implementation details and per-domain results, please refer to the Appendix C.

Among all the DG methods, SWAD performs well across all datasets with strong OOD scores, MIRO+SWAD combination improves results in general, particularly for Noise-Aware Generalization-

Table 1: **Results on real-world datasets**. Six groups of methods are presented: baseline (*ERM (Gulrajani & Lopez-Paz, 2020)*), DG methods (*SWAD (Cha et al., 2021), MIRO (Cha et al., 2022), ERM++ (Teterwak et al., 2023), SAGM (Wang et al., 2023)*), Robust-OOD methods (*VREx (Krueger et al., 2021), Fishr (Rame et al., 2022)*), Domain-aware optimization method (*DISAM (Zhang et al., 2024)*), LNL methods (*ELR (Liu et al., 2020), UNICON (Karim et al., 2022), DISC (Li et al., 2023), PLM (Zhao et al., 2024)*), and LNL+DG combination methods. For each dataset, both ID and OOD performance are reported. The combination methods show promising results in both ID and OOD tasks. Refer to Sec. 3.3 for more discussions.

| Method | Group | VLCS | | NAG-Fashion | | CHAMMI-CP | |
|---|---|---|---|---|---|---|---|
| | | ID | OOD | ID | OOD | ID | OOD |
| ERM | Baseline | 83.97 | 77.10 | 87.00 | 33.11 | 79.22 | 41.08 |
| SWAD | DG | **86.93** | 79.07 | 90.62 | 59.10 | 73.91 | 43.66 |
| MIRO | DG | 85.96 | 77.06 | 90.91 | 54.10 | 65.47 | **46.55** |
| ERM++ | DG | 79.15 | 77.68 | 83.30 | 38.22 | 72.49 | 44.55 |
| SAGM | DG | 86.78 | 78.75 | 91.85 | 34.40 | 77.11 | 41.19 |
| MIRO+SWAD | DG | 86.83 | 77.86 | 91.02 | 60.87 | 67.31 | 45.82 |
| SAGM+SWAD | DG | 86.63 | 79.41 | 91.43 | 38.59 | 78.27 | 41.45 |
| VREx | Robust-OOD | 83.65 | 76.02 | 87.10 | 49.92 | 74.78 | 44.81 |
| Fishr | Robust-OOD | 84.50 | 75.85 | 86.51 | 41.90 | 73.90 | 44.03 |
| DISAM | Domain Opt | 84.40 | 77.23 | 87.92 | 48.87 | 72.36 | 44.83 |
| ELR | LNL | 86.31 | 76.16 | 87.40 | 35.15 | 82.63 | 43.63 |
| UNICON | LNL | 84.85 | 77.39 | 87.31 | 53.85 | 76.72 | 42.02 |
| DISC | LNL | 83.79 | 76.65 | 87.25 | 47.01 | 43.28 | 41.28 |
| PLM | LNL | 82.85 | 75.60 | 87.43 | 27.06 | 70.47 | 44.44 |
| ERM++ + ELR | NAG | 84.83 | 78.11 | 83.73 | 35.84 | 75.72 | 42.04 |
| MIRO+UNICON | NAG | 84.95 | 76.21 | 87.74 | 52.98 | **84.52** | 43.44 |
| MIRO+SWAD+UNICON | NAG | 83.82 | 76.73 | 86.09 | 57.18 | 76.17 | 45.65 |
| MIRO+ELR | NAG | 85.04 | 77.51 | 91.11 | 31.52 | 74.54 | 41.28 |
| SWAD+ELR | NAG | 86.84 | **80.01** | 91.19 | 59.08 | 73.49 | 44.66 |
| MIRO+SWAD+ELR | NAG | 86.78 | 79.86 | **91.48** | **63.53** | 70.73 | 44.82 |

Fashion. For all the LNL methods, ELR performs consistently well. For the combination methods, SWAD+ELR shows the best OOD performance in VLCS. MIRO+SWAD+ELR achieves the highest scores in Noise-Aware Generalization-Fashion.

Methods combining multiple strategies (e.g., MIRO, SWAD, and ELR) generally perform better, especially in challenging OOD scenarios. Simple ERM struggles with OOD performance, highlighting the need for advanced techniques in handling domain generalization and noisy labels. Regularization techniques (ELR) and domain generalization methods (SWAD, MIRO) are effective in improving robustness across datasets.

Moreover, there are some **unexpected outcomes**. First, the ranking of LNL methods differs from other LNL benchmark datasets. Although UNICON is a newer state-of-the-art method and is expected to outperform ELR, its in-domain performance is consistently lower in the Noise-Aware Generalization benchmarks. Second, combining methods might negatively impact performance, as seen with the MIRO and UNICON+MIRO combination. We delve into these unusual results in Sec. 3.3.

## 3.3 ANALYSIS

Since NAG is a composite task that integrates two interrelated challenges, LNL and DG, our analysis begins by examining how introducing another factor affects the traditional task. Specifically, we investigate: How does multi-distribution data impact LNL methods? and How does noisy data impact DG methods? These questions address the core issue of **why NAG cannot be effectively solved using a single LNL or DG method alone**. Following this, we explore LNL and DG's interaction by

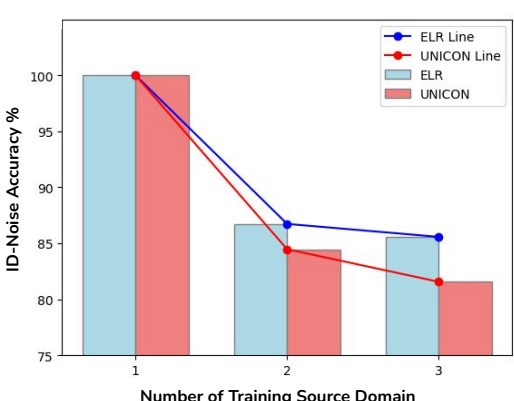

Figure 3: **ID accuracy comparisons for LNL methods when training with varying numbers of source domains on the VLCS** (Fang et al., 2013). "1 domain" refers to training on Caltech101 (Fei-Fei et al., 2004). "2 domains" is the average accuracy when training on Caltech101 plus one other domain from [LabelMe (Russell et al., 2008), VOC2007 (Everingham et al., 2010), SUN09 (Choi et al., 2010)]. "3 domains" is the average accuracy when training on Caltech101 plus two domains from the same set. ELR (Liu et al., 2020) consistently outperforms UNICON (Karim et al., 2022), with the performance gap widening as the number of training domains increases. See Sec. 3.3.1 for more discussions.

examining the trade-off between prioritizing cleaner samples or maintaining balanced distributions. Finally, we conclude by offering insights and recommendations for addressing NAG effectively.

### 3.3.1 HOW DOES MULTI-DISTRIBUTION TRAINING DATA IMPACT LNL METHODS?

*The performance of LNL methods declines when additional data sources with diverse distributions are introduced, with sample-selection methods being particularly impacted.* Fig. 3 shows the ID performance when training with varying numbers of source domains. Starting with a single, relatively simple domain like Caltech101 (Fei-Fei et al., 2004), the ID accuracy approaches nearly 100%. However, as the number of training domains increases, the task becomes more challenging for the model, leading to a decline in ID performance across all methods.

A key observation from the figure is the widening performance gap between ELR (Liu et al., 2020) and UNICON (Karim et al., 2022) as the number of training domains increases, **contrary to their ranking on other LNL datasets** (Karim et al., 2022). This suggests that sample selection methods like UNICON struggle more with noisy data when domain shifts are present. Specifically, it becomes increasingly difficult for UNICON to distinguish between samples from minority distributions and noisy samples as the diversity of the training data grows. This challenge is evident in Fig.4, where domains with fewer samples are selected less frequently. For instance, in the "person" class, the representation of Caltech data decreases significantly from 25.87% to 11.92% in the selected samples. In contrast, ELR maintains a relatively better performance, indicating its robustness in handling the complexities introduced by multiple, noisy domains.

### 3.3.2 HOW DOES NOISY TRAINING DATA IMPACT DG METHODS?

*Performance across domains shows varying levels of decline under noisy conditions, highlighting the sensitivity of DG methods to noise. SWAD+MIRO demonstrates exceptional resilience to noise.* Fig. 5-(a) shows the noise-sensitivity on different domains in DomainNet-SN (Peng et al., 2019) dataset. This variability means that methods effective in one domain may not necessarily perform well in another, underscoring the need for adaptable approaches that can handle diverse conditions. Fig. 5-(b) shows the comparison of DG methods on DomainNet with different degrees of asymmetric noise. The first observation is a consistent outperformance of SWAD over MIRO, which implies that SWAD's strategy of utilizing averages exhibits greater robustness compared to MIRO. Another point to highlight is the increasing performance gap as the noise ratio increases. This suggests that SWAD's robustness is advantageous when noise ratios are high.

**Noise sample selection skews domain distribution.** In Fig. 4, the sample selection process has substantially modified the original domain distribution. Models trained on this altered sample distribution might tend to overfit to the more prominently represented domains while potentially underperforming on the less represented ones. Consequently, DG methods striving for generalization across domains might encounter diminished effectiveness due to the disproportionate representation of domains in the training data. The difference between the original and selected-sample distributions highlights the importance of considering domain balance during sample selection.

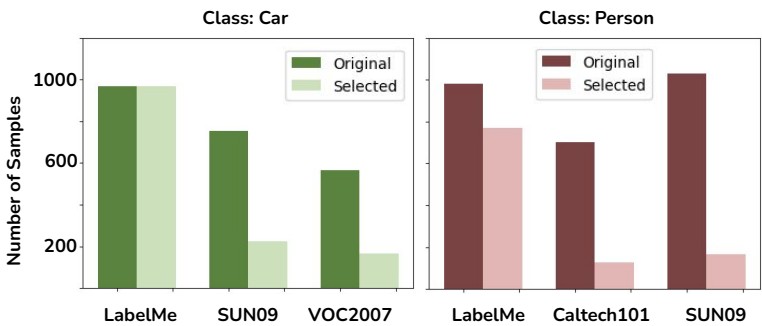

Figure 4: **Changes in domain distribution after the UNICON sample selection process on VLCS** (Fang et al., 2013). (*Left bar: number of samples before selection, right bar: after selection.*) These two cases illustrate a risk of skewing domain distributions from the LNL selection process. See Sec. 3.3.4 for more details.

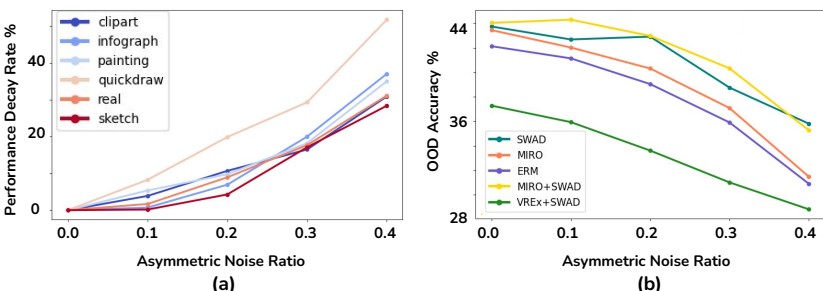

Figure 5: **OOD accuracy comparisons for DG methods with synthesized asymmetric noise on DomainNet-SN** (Peng et al., 2019). (a) Different domains exhibit varying sensitivity to asymmetric noise. The plot shows the degree of decrease in OOD performance for ERM applied to six domains as the noise ratio increases. (b) Comparisons of DG methods with increasing noise ratios. The results demonstrate that noise negatively impacts performance, but SWAD is more robust than MIRO and ERM, with a noticeable performance gap and SWAD+MIRO shows the best resilience to noise. Refer to Sec. 3.3.2 for more details.

### 3.3.3 CLEANER VS. BALANCED: WHICH ENHANCES ID AND OOD PERFORMANCE?

*Quality outweighs quantity in enhancing robustness.* As highlighted in earlier sections, imbalanced domain distributions pose additional challenges for LNL methods, while noise introduces difficulties for DG methods. This raises the question: how can we strike a balance between cleanliness and distributional balance?

Fig. 6 illustrates the relationship between domain balance, clean sample count, and ID/OOD performance in experiments on the "person" class from the VLCS dataset, which includes real-world noise. As labels have been manually verified, the total and clean sample distributions across four domains are known. The "person" class is chosen due to the originally balanced data distribution, despite varying numbers of clean samples.

In Fig. 6 (c), the x-axis shows the percentage of selected samples, using the JSD metric from UNICON (Karim et al., 2022) to identify the top $r$ samples with minimal JSD distance as "clean." The left y-axis shows the total sample count (*dark bars: true clean samples; light bars: false clean*), while the right y-axis shows ID and OOD accuracy.

At lower selection ratios ($r$), the distribution becomes less balanced, as more samples are drawn from the cleaner VOC2007 domain. At higher ratios, balance is maintained but with increased noise. Results in (c) indicate that OOD performance isn't improved by merely balancing distributions; added noise reduces accuracy. With the best results at r = 0.2, suggesting **"quality"** is more crucial than **"quantity"** for robustness enhancement.

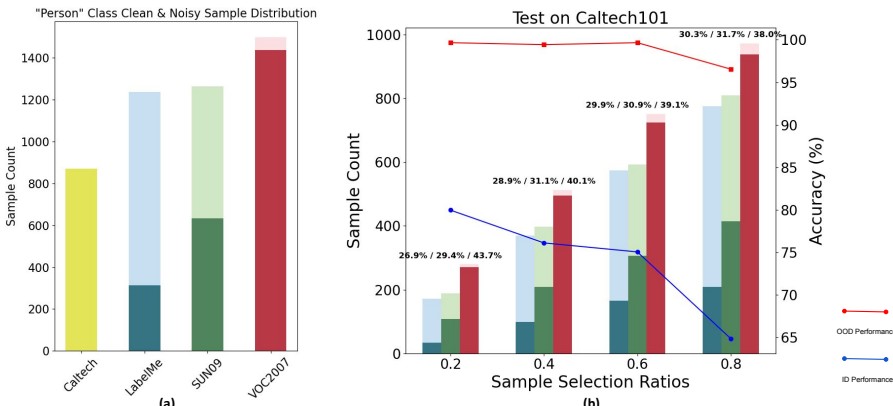

Figure 6: **Balance, Number of Clean Samples, and ID/OOD Performance on the VLCS Dataset "Person" Class**. (a) Sample distribution across four domains (*dark color: clean, light color: noisy*). While total sample counts are similar, clean sample counts vary. (b) Testing on the Caltech101 domain with training on the remaining domains. The x-axis shows the variation in sample selection ratio per class, while the ratios for each domain are shown at the top of the bars. The observed decrease in both ID and OOD performance, as the distribution becomes more balanced and sample size increases, suggests that a more balanced distribution does not necessarily enhance OOD accuracy and that increased noise adversely affects both ID and OOD performance. See Sec. 3.3.3 for discussions.

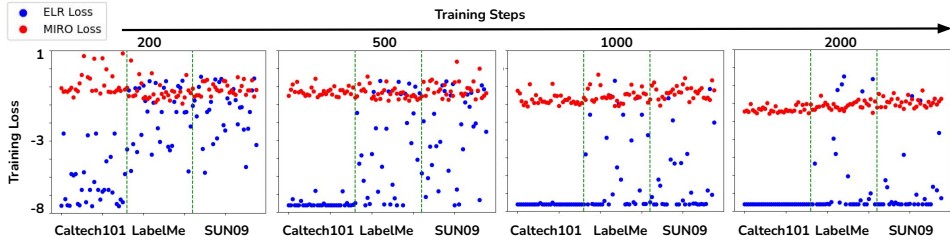

Figure 7: **Changes in LNL and DG losses over time on VLCS** (Fang et al., 2013). Each subplot represents a time step in the training process, divided into three blocks showing samples from three specific domains. MIRO maintains a consistent range across the domains, whereas ELR demonstrates a clear convergence pattern for different domains. Refer to Sec. 3.3.4 for more details.

### 3.3.4   WHAT ARE THE INSIGHTS FOR COMBINING LNL AND DG METHODS?

**Regularization-based techniques are more effective.** Table 1 shows an interesting pattern: datasets where domain shifts are more significant (VLCS and NAG-Fashion) regularization-based methods from the DG literature are generally more effective, whereas on CHAMMI-CP where label noise is more of an issue, LNL regularization is more effective (*e.g.*, ELR). Combining these generally improves performance. Other LNL methods that try to correct labels, *e.g.*, UNICON, can be effective in the low domain shift setting when combined with regularization techniques Table 1 or when domain labels are available to minimize domain shift Table 2, discussed below.

**Combining with disjoint losses.** Fig. 7 contains scatter plots tracking the loss over training steps for two different loss functions: ELR (Liu et al., 2020) (blue dots) and MIRO (Cha et al., 2022) (red dots). Each subplot represents the distribution of loss values at a specific training step: 200, 500, 1000, and 5000. The green dotted lines separate the three training domains at each step: Caltech101 (Fei-Fei et al., 2004) (left), LabelMe (Russell et al., 2008) (middle), SUN09 (Choi et al., 2010) (right).

ELR forms two distinct loss groups during training: one with low loss, indicating strong convergence on certain batches, and another fluctuating near zero, reflecting challenging or under-learned batches. Caltech101 reaches low loss first, aligning with its higher test accuracy, highlighting ELR's efficiency in learning specific data. In contrast, MIRO shows steadier but slower convergence, demonstrating stability and robustness across batches. This figure showcases how ELR and MIRO's complementary behaviors can enhance performance when combined.

Table 2: **Results of training with domain labels**. Adding domain labels to preserve the distribution during sample selection shows promising enhancements. Refer to Sec. 3.3.4 for details.

| Method | VLCS Fang et al. (2013) | | CHAMMI-CP Chen et al. (2024b) | |
|---|---|---|---|---|
| | ID | OOD | ID | OOD |
| UNICON (Karim et al., 2022) | 84.85 | 77.39 | 76.72 | 42.02 |
| UNICON + *domain label* | **85.78** | **78.16** | **77.60** | **43.37** |
| MIRO+UNICON (Cha et al., 2022) | 84.95 | 76.21 | **84.52** | 43.44 |
| MIRO+UNICON+ *domain label* | **86.00** | **78.57** | 78.44 | **45.24** |
| MIRO+SWAD+UNICON (Cha et al., 2021) | 83.82 | 76.73 | 76.17 | **45.65** |
| MIRO+SWAD+UNICON+ *domain label* | **85.63** | **78.26** | **76.49** | 43.56 |

**Adapting LNL methods when domain labels are available.** Tab. 2 presents the results of training with domain labels on the VLCS and CHAMMI-CP datasets, where the domain labels are available. The evaluation metrics include performance on in-domain noise (ID) and out-of-domain (OOD) data. As discussed above, the sample selection may skew the domain distribution, so the following results show our exploration of whether utilizing the domain label to maintain the domain distribution would be beneficial. the methods compared in the table are UNICON, MIRO+UNICON, and MIRO+SWAD+UNICON, both with and without the inclusion of domain labels. For methods with domain labels, clean samples are selected per class and per domain.

Adding domain labels for the LNL SOTA method UNICON improved both ID and OOD data performance for both datasets. For MIRO+UNICON and MIRO+SWAD+UNICON, adding domain labels enhanced performance on both metrics on VLCS dataset. The inclusion of domain labels generally improves model performance, indicating that domain-specific information can enhance robustness and generalization.

## 4 CONCLUSION AND DISCUSSION

This work tackles the challenges of noisy, diverse real-world data by introducing Noise-Aware Generalization, a task focused on managing in-domain noise and out-of-domain generalization. We propose a unified framework combining Learning with Noisy Labels (LNL) and Domain Generalization (DG) approaches, supported by comprehensive experiments on three real-world datasets with varying noise ratios and domain shifts.Our evaluation included state-of-the-art methods from both LNL and DG fields, as well as their combinations. Surprisingly, no single method consistently outperformed others, showing the complexity of this problem. Key insights from our work include: LNL methods struggle to differentiate noise from diverse distributions, and their sample selection can distort domain distributions, harming OOD performance. Prioritizing quality over quantity enhances robustness in Noise-Aware Generalization .

We provide the following specific recommendations based on our experiments. Generally there are two components to any dataset: the amount of noise and the strength of the domain shift. The inherent inability to separate these two factors mean that regularization-based techniques (*e.g.*, SWAD and MIRO) are more effective. From here, our recommendations diverge based on the amount of domain shift present. In cases of high domain shift (*e.g.*, VLCS and NAG-Fashion), domain labels are required to use other techniques (*e.g.*, pseudo-labeling UNICON for addressing noisy labels), as they the effect of domain shifts are minimized. If the domains during training are smaller than those seen at test time (*e.g.*, NAG-Fashion), then additional regularization may be required. In cases of low domain shift (*e.g.*, CHAMMI-CP) combining regularization with other techniques like UNICON can be used immediately even in cases of high noise, but too much regularization can be detrimental.

**Limitations.** Our focus on in-domain noise has mainly involved closed-set noise. Future research could explore Noise-Aware Generalization in the context of open-set noise, prevalent in real-world datasets like those from web crawling.

# 5 ETHICS STATEMENT

This paper addresses a unified task that requires models to perform well on both in-domain and out-of-domain data when training on datasets with label noise. This can result in models that can effectively learn from a wide variety of data, including cell painting data where prior work in tasks like LNL found especially challenging due to its high amounts of label noise that is useful as a step towards drug discovery (Wang & Plummer, 2024). However, like our topics in this field, also can enable bad actors to use these models to train more effective recognition systems for nefarious purposes. Additionally, users should be mindful that although we provide an evaluation on a diverse set of datasets, they still make mistakes in their predictions that may vary depending on the dataset. Thus, researchers and engineers should be mindful of these factors when deploying a system for end-users.

# 6 REPRODUCIBILITY STATEMENT

We will release our code to ensure it can be reproduced upon acceptance. This will include code for training/testing the models we compared to in a unified codebase where additional methods can be easily integrated and the data loaders required to evaluate models on our benchmarks. We will also include pretrained models for ease of use.

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

## A    DOMAINNET WITH SYNTHESIZED NOISE (DOMAINNET-SN)

To control the noise ratio and add variety to the benchmark datasets, DomainNet with 345 classes is augmented with synthesized asymmetric noise. Unlike symmetric noise, where noise is uniformly sampled from all other classes, asymmetric noise is sampled from specific classes. In our setting, each class has a single noise source class. For example, as shown in Table 4, for class index 0, the noise source is class index 308. If the noise ratio is set to be $p$, it means a sample has a probability of $p$ to flip to the noisy label 308.

The asymmetric noise pairs are determined using the validation confusion matrix. We select 20% of the samples as the validation set and the rest are used for training. After training for one epoch with ERM (Gulrajani & Lopez-Paz, 2020), we generate the confusion matrix for the validation set. For each class, the class with the highest number of predictions (excluding its own class) is selected as the noise source.

## B    VLCS NOISE

## C    EXPERIMENTS

This section presents the experimental details including model architecture, algorithm implementation, hyperparameter choices, etc. We provide the code in a zip file along with this supplementary and will open-source the code upon acceptance.

### C.1    MODEL ARCHITECTURE

For the VLCS, DomainNet-SN, and Robust-Fashion datasets, we used ResNet50 (He et al., 2016) model pretrained on ImageNet (Deng et al., 2009) as the foundational architecture. Conversely, for the CHAMMI-CP dataset, we follow the architecture outlined in the benchmark paper (Chen et al., 2024b), employing a ConvNeXt (Liu et al., 2022b) model pretrained on ImageNet 22K (Deng et al., 2009) as the backbone. To accommodate the CP images with five input channels, we made necessary adjustments to the first input layer.

### C.2    INTEGRATED METHODS

Algorithm 1, 2, 3, 4, 5, 6 show the detail of the integrated methods.

---

**Algorithm 1:** ERM++ + ELR Algorithm.

---

**Input** : Sample inputs $X = \{x_i\}_{i=1}^n$, noisy labels $\widetilde{Y} = \{\widetilde{y_i}\}_{i=1}^n$, ELR temporal ensembling momentum $\beta$, regularization parameter $\lambda$, neural network with trainable parameters $f_\theta$

**Output** : Neural network with updated parameters $f_{\theta'}$

**for** $step \leftarrow 1$ **to** $training\_steps$ **do**
    **for** *minibatch B* **do**
        **for** *i in B* **do**
            $p_i = f_\theta(x_i)$; // Model prediction.
            $t_i = \beta * t_i + (1 - \beta) * p_i$; // Temporal ensembling.
        **end**
        loss $= -\frac{1}{|B|}\Sigma_{|B|}cross\_entropy(p_i, y_i) + \frac{\lambda}{|B|}\Sigma_{|B|}log(1- <p_i, t_i>)$; // ELR loss: cross entropy loss and regularization loss.
        Update $f_\theta$.
    **end**
    $f_{\theta'}$ = Update $f_\theta$ with ERM++ parameter averaging.
**end**

---

---

**Algorithm 2:** MIRO + ELR Algorithm.

---

**Input** : Sample inputs $X = \{x_i\}_{i=1}^n$, noisy labels $\widetilde{Y} = \{\widetilde{y_i}\}_{i=1}^n$, ELR temporal ensembling momentum $\beta$, ELR regularization parameter $\lambda1$, MIRO regularization parameter $\lambda2$, MIRO mean encoder $\mu$, MIRO variance encode $\sigma$, feature extractor with trainable parameters $f_\theta$, pretrained feature extractor with parameters $f_{\theta_0}$

**Output** : Neural network with updated parameters $f_{\theta'}$

**for** $step \leftarrow 1$ **to** $training\_steps$ **do**
    **for** *minibatch B* **do**
        **for** *i in B* **do**
            $p_i = f_\theta(x_i)$; // feature extractor output.
            $p_i^0 = f_{\theta_0}(x_i)$; // Pretrained feature extractor output.
            $t_i = \beta * t_i + (1 - \beta) * p_i$; // Temporal ensembling.
        **end**
        loss $= -\frac{1}{|B|}\Sigma_{|B|}cross\_entropy(p_i, y_i)$; // Cross entropy loss.
        loss $+= \frac{\lambda1}{|B|}\Sigma_{|B|}log(1- <p_i, t_i>)$; // ELR loss with regularization term.
        loss $+= \frac{\lambda2}{|B|}\Sigma_{|B|}(log(|\sigma(p_i)|) + ||p_i^0 - \mu(p_i)||^2_{\sigma(p_i)^{-1}})$; // MIRO loss with regularization term.
        Update $f_\theta$.
    **end**
    $f_{\theta'}$ = Updated $f_\theta$.
**end**

---

**Algorithm 3:** SWAD + ELR Algorithm.

---

**Input** : Sample inputs $X = \{x_i\}_{i=1}^n$, noisy labels $\widetilde{Y} = \{\widetilde{y_i}\}_{i=1}^n$, ELR temporal ensembling momentum $\beta$, ELR regularization parameter $\lambda$, neural network with trainable parameters $f_\theta$

**Output** : Neural network with updated parameters $f_{\theta'}$

**for** $step \leftarrow 1$ **to** $training\_steps$ **do**
    **for** *minibatch B* **do**
        **for** *i in B* **do**
            $p_i = f_\theta(x_i)$ ; // Model prediction.
            $t_i = \beta * t_i + (1 - \beta) * p_i$ ; // Temporal ensembling.
        **end**
        loss $= -\frac{1}{|B|}\Sigma_{|B|}cross\_entropy(p_i, y_i) + \frac{\lambda}{|B|}\Sigma_{|B|}log(1- <p_i, t_i>)$ ; // ELR loss: cross entropy loss and regularization loss.
        Update $f_\theta$. Decide the start $step_s$ and end $step_e$ iteration for averaging in SWAD.
    **end**
    $f_{\theta'} = \frac{1}{step_e - step_s + 1}\Sigma f_\theta$ ; // SWAD parameter averaging.
**end**

---

**Algorithm 4:** MIRO + SWAD + ELR Algorithm.

---

**Input** : Sample inputs $X = \{x_i\}_{i=1}^n$, noisy labels $\widetilde{Y} = \{\widetilde{y_i}\}_{i=1}^n$, ELR temporal ensembling momentum $\beta$, ELR regularization parameter $\lambda 1$, MIRO regularization parameter $\lambda 2$, MIRO mean encoder $\mu$, MIRO variance encode $\sigma$, feature extractor with trainable parameters $f_\theta$, pretrained feature extractor with parameters $f_{\theta_0}$

**Output** : Neural network with updated parameters $f_{\theta'}$

**for** $step \leftarrow 1$ **to** $training\_steps$ **do**
    **for** *minibatch B* **do**
        **for** *i in B* **do**
            $p_i = f_\theta(x_i)$ ; // feature extractor output.
            $p_i^0 = f_{\theta_0}(x_i)$ ; // Pretrained feature extractor output.
            $t_i = \beta * t_i + (1 - \beta) * p_i$ ; // Temporal ensembling.
        **end**
        loss $= -\frac{1}{|B|}\Sigma_{|B|}cross\_entropy(p_i, y_i)$ ; // Cross entropy loss.
        loss $+=\frac{\lambda 1}{|B|}\Sigma_{|B|}log(1- <p_i, t_i>)$ ; // ELR loss with regularization term.
        loss $+= \frac{\lambda 2}{|B|}\Sigma_{|B|}(log(|\sigma(p_i)|) + ||p_i^0 - \mu(p_i)||^2_{\sigma(p_i)^{-1}})$ ; // MIRO loss with regularization term.
        Update $f_\theta$. Decide the start $step_s$ and end $step_e$ iteration for averaging in SWAD.
    **end**
    $f_{\theta'} = \frac{1}{step_e - step_s + 1}\Sigma f_\theta$ ; // SWAD parameter averaging.
**end**

---

**Algorithm 5:** MIRO + UNICON Algorithm.

---

**Input** : Sample inputs $X = \{x_i\}_{i=1}^n$, noisy labels $\widetilde{Y} = \{\widetilde{y_i}\}_{i=1}^n$, MIRO regularization parameter $\lambda 2$, MIRO mean encoder $\mu$, MIRO variance encode $\sigma$, feature extractor-1 with trainable parameters $f1_\theta$, feature extractor-2 with trainable parameters $f2_\theta$, pretrained feature extractor with parameters $f_{\theta_0}$, UNICON sharpening temperature $T$, UNICON unsupervised loss coefficient $\lambda_u$, UNICON contrastive loss coefficient $\lambda_c$, , UNICON regularization loss coefficient $\lambda_r$.

**Output :** Neural network with updated parameters $f1_{\theta'}$ and $f2_{\theta'}$

**for** $step \leftarrow 1$ **to** $training\_steps$ **do**

  $D_{clean}, D_{noisy} = UNICON - Selection(X = \{x_i\}_{i=1}^n, f1_\theta, f2_\theta),$ ; // `UNICON clean-noisy sample selection.`

  **for** *clean minibatch $B_{clean}$* **do**

    **for** *noisy minibatch $B_{noisy}$* **do**

      **for** *$i$ in $B = B_{clean} \bigcup B_{noisy}$* **do**

        $p1_i = f1_\theta(x_i)$ ; // `feature extractor-1 output.`

        $p2_i = f2_\theta(x_i)$ ; // `feature extractor-2 output.`

        $p_i^0 = f_{\theta_0}(x_i)$ ; // `Pretrained feature extractor output.`

      **end**

      $loss_1 = -\frac{1}{|B|}\Sigma_{|B|}cross\_entropy(p1_i, y_i)$ ; // `Cross entropy loss for feature extractor-1.`

      $loss_1 += \frac{\lambda 2}{|B|}\Sigma_{|B|}(log(|\sigma(p1_i)|) + ||p_i^0 - \mu(p1_i)||_{\sigma(p1_i)^{-1}}^2)$ ; // `MIRO loss with regularization term for feature extractor-1.`

      $loss_2 = -\frac{1}{|B|}\Sigma_{|B|}cross\_entropy(p2_i, y_i)$ ; // `Cross entropy loss for feature extractor-2.`

      $loss_2 += \frac{\lambda 2}{|B|}\Sigma_{|B|}(log(|\sigma(p2_i)|) + ||p_i^0 - \mu(p2_i)||_{\sigma(p2_i)^{-1}}^2)$ ; // `MIRO loss with regularization term for feature extractor-2.`

      $X_{clean|B|}^{weak}$ = weak-augmentation($B_{clean}$)

      $X_{noisy|B|}^{weak}$ = weak-augmentation($B_{noisy}$)

      $X_{clean|B|}^{strong}$ = strong-augmentation($B_{clean}$)

      $X_{noisy|B|}^{strong}$ = strong-augmentation($B_{noisy}$)

      Get labeled set with UNICON label refinement on clean batch.

      Get unlabeled set with UNICON pseudo label on noisy batch.

      $L_{u1}, L_{u2}$ = MixMatch on labeled and unlabeled sets ; // `UNICON unsupervised loss for feature extractor-1 and extractor-2.`

      Get $L_{c1}, L_{c2}$ ; // `UNICON contrastive loss for feature extractor-1 and extractor-2.`

      Get $L_{r1}, L_{r2}$ ; // `UNICON regularization loss for feature extractor-1 and extractor-2.`

      $loss_1 += \lambda_u * L_{u1} + \lambda_c * L_{c1} + \lambda_r * L_{r1}$ ; // `Update UNICON loss for feature extractor-1.`

      $loss_2 += \lambda_u * L_{u2} + \lambda_c * L_{c2} + \lambda_r * L_{r2}$ ; // `Update UNICON loss for feature extractor-2.`

      Update $f1_\theta$ and $f2_\theta$.

    **end**

  **end**

  $f1_{\theta'}$ = Updated $f1_\theta$, $f2_{\theta'}$ = Updated $f2_\theta$.

**end**

---

**Algorithm 6:** MIRO + SWAD + UNICON Algorithm.

**Input** : Sample inputs $X = \{x_i\}_{i=1}^n$, noisy labels $\widetilde{Y} = \{\widetilde{y_i}\}_{i=1}^n$, MIRO regularization parameter $\lambda 2$, MIRO mean encoder $\mu$, MIRO variance encode $\sigma$, feature extractor-1 with trainable parameters $f1_\theta$, feature extractor-2 with trainable parameters $f2_\theta$, pretrained feature extractor with parameters $f_{\theta_0}$, UNICON sharpening temperature $T$, UNICON unsupervised loss coefficient $\lambda_u$, UNICON contrastive loss coefficient $\lambda_c$, , UNICON regularization loss coefficient $\lambda_r$.

**Output** : Neural network with updated parameters $f1_{\theta'}$ and $f2_{\theta'}$

**for** $step \leftarrow 1$ **to** $training\_steps$ **do**

    $D_{clean}, D_{noisy} = UNICON - Selection(X = \{x_i\}_{i=1}^n, f1_\theta, f2_\theta), ;$ // UNICON clean-noisy sample selection.

    **for** *clean minibatch* $B_{clean}$ **do**

        **for** *noisy minibatch* $B_{noisy}$ **do**

            **for** $i$ *in* $B = B_{clean} \bigcup B_{noisy}$ **do**

                $p1_i = f1_\theta(x_i) ;$ // feature extractor-1 output.

                $p2_i = f2_\theta(x_i) ;$ // feature extractor-2 output.

                $p_i^0 = f_{\theta_0}(x_i) ;$ // Pretrained feature extractor output.

            **end**

            $loss_1 = -\frac{1}{|B|}\Sigma_{|B|}cross\_entropy(p1_i, y_i) ;$ // Cross entropy loss for feature extractor-1.

            $loss_1 += \frac{\lambda 2}{|B|}\Sigma_{|B|}(log(|\sigma(p1_i)|) + ||p_i^0 - \mu(p1_i)||^2_{\sigma(p1_i)^{-1}}) ;$ // MIRO loss with regularization term for feature extractor-1.

            $loss_2 = -\frac{1}{|B|}\Sigma_{|B|}cross\_entropy(p2_i, y_i) ;$ // Cross entropy loss for feature extractor-2.

            $loss_2 += \frac{\lambda 2}{|B|}\Sigma_{|B|}(log(|\sigma(p2_i)|) + ||p_i^0 - \mu(p2_i)||^2_{\sigma(p2_i)^{-1}}) ;$ // MIRO loss with regularization term for feature extractor-2.

            $X^{weak}_{clean|B|} = $ weak-augmentation$(B_{clean})$

            $X^{weak}_{noisy|B|} = $ weak-augmentation$(B_{noisy})$

            $X^{strong}_{clean|B|} = $ strong-augmentation$(B_{clean})$

            $X^{strong}_{noisy|B|} = $ strong-augmentation$(B_{noisy})$

            Get labeled set with UNICON label refinement on clean batch.

            Get unlabeled set with UNICON pseudo label on noisy batch.

            $L_{u1}, L_{u2} = $ MixMatch on labeled and unlabeled sets ; // UNICON unsupervised loss for feature extractor-1 and extractor-2.

            Get $L_{c1}, L_{c2}$ ; // UNICON contrastive loss for feature extractor-1 and extractor-2.

            Get $L_{r1}, L_{r2}$ ; // UNICON regularization loss for feature extractor-1 and extractor-2.

            $loss_1 += \lambda_u * L_{u1} + \lambda_c * L_{c1} + \lambda_r * L_{r1}$ ; // Update UNICON loss for feature extractor-1.

            $loss_2 += \lambda_u * L_{u2} + \lambda_c * L_{c2} + \lambda_r * L_{r2}$ ; // Update UNICON loss for feature extractor-2.

            Update $f1_\theta$ and $f2_\theta$. Decide the start $step_s$ and end $step_e$ iteration for averaging in SWAD.

        **end**

    **end**

    $f1_{\theta'} = \frac{1}{step_e - step_s + 1}\Sigma f1_\theta$ $f2_{\theta'} = \frac{1}{step_e - step_s + 1}\Sigma f2_\theta$ ; // SWAD parameter averaging.

**end**

Table 3: VLCS Dataset Overview (Total Samples, Noisy Samples)

| Domain | Category | Total Samples | Noisy Samples |
|---|---|---|---|
| Caltech | Bird | 237 | 1 (with person) |
| | Car | 123 | 0 (black & white car imgs) |
| | Chair | 118 | 0 |
| | Dog | 67 | 0 (only black and white dog) |
| | Person | 870 | 0 (profile photos with redundancy) |
| LabelMe | Bird | 80 | 20 |
| | Car | 1209 | 559 (background: building, road, mountains; small & incomplete cars, unclear night imgs [OOD]) |
| | Chair | 89 | 61 (over half have cars, person) |
| | Dog | 43 | 25 (with person, cars) |
| | Person | 1238 | 924 (over 80% noisy images have cars, street photos are similar to car and chair categories, small person figures) |
| SUN09 | Bird | 21 | 12 (background, 1 person and dog) |
| | Car | 933 | 548 (street view, buildings, person) |
| | Chair | 1036 | 186 (mostly person, very few car interior) |
| | Dog | 31 | 25 ($\sim$20 noisy images with person) |
| | Person | 1265 | 631 (very small person figures) |
| VOC2007 | Bird | 330 | 29 (mostly human, a few cars, one small bird) |
| | Car | 699 | 133 (mostly person, $\sim$5 don't have cars) |
| | Chair | 428 | 145 (mostly person, some cars, very few missing chair) |
| | Dog | 420 | 111 (mostly human, a few cars) |
| | Person | 1499 | 61 (mostly cars, some don't have person) |

## C.3 IMPLEMENTATION DETAILS

We incorporate the implementation of the ERM++ [1] (Teterwak et al., 2023), DISC [2] (Li et al., 2023), UNICON [3] (Karim et al., 2022), ELR [4] (Liu et al., 2020), SAGM [5] (Wang et al., 2023), MIRO [6] (Cha

---

[1] https://github.com/piotr-teterwak/erm_plusplus

[2] https://github.com/JackYFL/DISC

[3] https://github.com/nazmul-karim170/UNICON-Noisy-Label

[4] https://github.com/shengliu66/ELR

[5] https://github.com/Wang-pengfei/SAGM

[6] https://github.com/kakaobrain/miro

Table 4: Asymmetrical Noise Dictionary

| Key | Value | Key | Value | Key | Value | Key | Value | Key | Value | Key | Value |
|-----|-------|-----|-------|-----|-------|-----|-------|-----|-------|-----|-------|
| 0 | 308 | 1 | 208 | 2 | 28 | 3 | 135 | 4 | 5 | 5 | 0 |
| 6 | 0 | 7 | 324 | 8 | 324 | 9 | 208 | 10 | 288 | 11 | 324 |
| 12 | 208 | 13 | 285 | 14 | 208 | 15 | 16 | 16 | 17 | 17 | 282 |
| 18 | 19 | 19 | 327 | 20 | 309 | 21 | 208 | 22 | 327 | 23 | 208 |
| 24 | 288 | 25 | 135 | 26 | 27 | 27 | 28 | 28 | 208 | 29 | 208 |
| 30 | 327 | 31 | 98 | 32 | 33 | 33 | 144 | 34 | 35 | 35 | 308 |
| 36 | 282 | 37 | 38 | 38 | 327 | 39 | 208 | 40 | 208 | 41 | 42 |
| 42 | 208 | 43 | 44 | 44 | 308 | 45 | 46 | 46 | 331 | 47 | 324 |
| 48 | 91 | 49 | 90 | 50 | 327 | 51 | 324 | 52 | 53 | 53 | 324 |
| 54 | 327 | 55 | 331 | 56 | 282 | 57 | 151 | 58 | 334 | 59 | 324 |
| 60 | 324 | 61 | 208 | 62 | 175 | 63 | 64 | 64 | 327 | 65 | 208 |
| 66 | 67 | 67 | 68 | 68 | 208 | 69 | 208 | 70 | 138 | 71 | 331 |
| 72 | 324 | 73 | 175 | 74 | 53 | 75 | 254 | 76 | 338 | 77 | 276 |
| 78 | 91 | 79 | 208 | 80 | 282 | 81 | 208 | 82 | 282 | 83 | 319 |
| 84 | 85 | 85 | 208 | 86 | 310 | 87 | 324 | 88 | 208 | 89 | 90 |
| 90 | 91 | 91 | 208 | 92 | 323 | 93 | 285 | 94 | 95 | 95 | 261 |
| 96 | 276 | 97 | 98 | 98 | 324 | 99 | 282 | 100 | 288 | 101 | 102 |
| 102 | 103 | 103 | 327 | 104 | 110 | 105 | 288 | 106 | 107 | 107 | 282 |
| 108 | 276 | 109 | 110 | 110 | 324 | 111 | 110 | 112 | 288 | 113 | 114 |
| 114 | 157 | 115 | 208 | 116 | 327 | 117 | 98 | 118 | 327 | 119 | 208 |
| 120 | 208 | 121 | 110 | 122 | 324 | 123 | 208 | 124 | 125 | 125 | 208 |
| 126 | 208 | 127 | 324 | 128 | 129 | 129 | 208 | 130 | 327 | 131 | 208 |
| 132 | 208 | 133 | 28 | 134 | 135 | 135 | 136 | 136 | 324 | 137 | 138 |
| 138 | 35 | 139 | 282 | 140 | 324 | 141 | 208 | 142 | 208 | 143 | 282 |
| 144 | 324 | 145 | 146 | 146 | 282 | 147 | 148 | 148 | 208 | 149 | 208 |
| 150 | 151 | 151 | 98 | 152 | 153 | 153 | 308 | 154 | 208 | 155 | 341 |
| 156 | 157 | 157 | 208 | 158 | 324 | 159 | 208 | 160 | 208 | 161 | 98 |
| 162 | 163 | 163 | 208 | 164 | 282 | 165 | 308 | 166 | 230 | 167 | 1 |
| 168 | 285 | 169 | 208 | 170 | 171 | 171 | 208 | 172 | 208 | 173 | 208 |
| 174 | 175 | 175 | 208 | 176 | 282 | 177 | 178 | 178 | 110 | 179 | 246 |
| 180 | 208 | 181 | 282 | 182 | 324 | 183 | 282 | 184 | 208 | 185 | 324 |
| 186 | 324 | 187 | 188 | 188 | 282 | 189 | 190 | 190 | 324 | 191 | 282 |
| 192 | 193 | 193 | 135 | 194 | 35 | 195 | 28 | 196 | 282 | 197 | 307 |
| 198 | 178 | 199 | 208 | 200 | 208 | 201 | 28 | 202 | 324 | 203 | 282 |
| 204 | 208 | 205 | 206 | 206 | 282 | 207 | 208 | 208 | 91 | 209 | 324 |
| 210 | 211 | 211 | 212 | 212 | 213 | 213 | 288 | 214 | 208 | 215 | 216 |
| 216 | 282 | 217 | 246 | 218 | 335 | 219 | 276 | 220 | 282 | 221 | 222 |
| 222 | 208 | 223 | 327 | 224 | 110 | 225 | 285 | 226 | 208 | 227 | 228 |
| 228 | 208 | 229 | 324 | 230 | 327 | 231 | 232 | 232 | 208 | 233 | 282 |
| 234 | 282 | 235 | 324 | 236 | 327 | 237 | 208 | 238 | 285 | 239 | 240 |
| 240 | 331 | 241 | 285 | 242 | 324 | 243 | 208 | 244 | 309 | 245 | 107 |
| 246 | 247 | 247 | 248 | 248 | 324 | 249 | 321 | 250 | 251 | 251 | 288 |
| 252 | 135 | 253 | 254 | 254 | 327 | 255 | 208 | 256 | 208 | 257 | 341 |
| 258 | 208 | 259 | 135 | 260 | 261 | 261 | 262 | 262 | 208 | 263 | 213 |
| 264 | 208 | 265 | 327 | 266 | 208 | 267 | 268 | 268 | 269 | 269 | 208 |
| 270 | 309 | 271 | 208 | 272 | 273 | 273 | 135 | 274 | 208 | 275 | 276 |
| 276 | 277 | 277 | 324 | 278 | 279 | 279 | 208 | 280 | 281 | 281 | 282 |
| 282 | 282 | 283 | 208 | 284 | 285 | 285 | 98 | 286 | 282 | 287 | 208 |
| 288 | 310 | 289 | 324 | 290 | 282 | 291 | 309 | 292 | 208 | 293 | 294 |
| 294 | 208 | 295 | 324 | 296 | 327 | 297 | 208 | 298 | 208 | 299 | 324 |
| 300 | 208 | 301 | 285 | 302 | 324 | 303 | 282 | 304 | 282 | 305 | 282 |
| 306 | 307 | 307 | 308 | 308 | 282 | 309 | 282 | 310 | 341 | 311 | 208 |
| 312 | 313 | 313 | 331 | 314 | 282 | 315 | 282 | 316 | 282 | 317 | 282 |
| 318 | 282 | 319 | 327 | 320 | 327 | 321 | 282 | 322 | 208 | 323 | 324 |
| 324 | 325 | 325 | 324 | 326 | 327 | 327 | 282 | 328 | 329 | 329 | 282 |
| 330 | 282 | 331 | 332 | 332 | 324 | 333 | 282 | 334 | 335 | 335 | 208 |
| 336 | 337 | 337 | 338 | 338 | 208 | 339 | 340 | 340 | 341 | 341 | 342 |
| 342 | 208 | 343 | 344 | 344 | 282 | | | | | | |

Table 5: **Learning rate** on VLCS (Fang et al., 2013),Noise-Aware Generalization-Fashion (Xiao et al., 2015; 2017) ,CHAMMI-CP (Chen et al., 2024b) and DomainNet-SN. Six groups of methods are presented: baseline (*ERM (Gulrajani & Lopez-Paz, 2020)*), DG methods (*SWAD (Cha et al., 2021), MIRO (Cha et al., 2022), ERM++ (Teterwak et al., 2023), SAGM (Wang et al., 2023)*), Robust-OOD methods (*VREx (Krueger et al., 2021), Fishr (Rame et al., 2022)*), Domain-aware optimization method (*DISAM (Zhang et al., 2024)*), LNL methods (*ELR (Liu et al., 2020), UNICON (Karim et al., 2022), DISC (Li et al., 2023), PLM (Zhao et al., 2024)*), and LNL+DG combination methods.

| Method | VLCS | Noise-Aware Generalization-Fashion | CHAMMI-CP | DomainNet-SN |
|---|---|---|---|---|
| ERM | 1e-3 | 1e-3 | 5e-5 | 1e-3 |
| ERM++ | 5e-5 | 1e-3 | 5e-5 | - |
| MIRO | 5e-5 | 5e-5 | 5e-5 | 5e-5 |
| SWAD | 5e-5 | 5e-5 | 5e-5 | 5e-5 |
| MIRO+SWAD | 5e-5 | 5e-5 | 5e-5 | - |
| SAGM | 3e-5 | 3e-5 | 1e-4 | - |
| SAGM+SWAD | 3e-5 | 3e-5 | 1e-4 | - |
| Fishr | 5e-5 | 5e-5 | 5e-5 | - |
| VREx | 5e-5 | 5e-5 | 5e-5 | - |
| DISAM | 5e-5 | 5e-5 | 5e-5 | - |
| ELR | 1e-3 | 1e-3 | 5e-5 | - |
| DISC | 1e-3 | 1e-3 | 5e-5 | - |
| UNICON | 5e-3 | 5e-3 | 5e-5 | - |
| PLM | 1e-3 | 5e-5 | 5e-5 | - |
| ERM++ + ELR | 5e-5 | 1e-3 | 5e-5 | - |
| MIRO+UNICON | 5e-5 | 5e-5 | 5e-5 | - |
| MIRO+SWAD+UNICON | 5e-5 | 5e-5 | 5e-5 | - |
| MIRO+ELR | 5e-5 | 5e-5 | 5e-5 | - |
| SWAD+ELR | 5e-5 | 5e-5 | 5e-5 | - |
| MIRO+SWAD+ELR | 5e-5 | 5e-5 | 5e-5 | - |

et al., 2022), VREx [7] (Krueger et al., 2021), Fishr [8] (Rame et al., 2022), DISAM [9] (Zhang et al., 2024), PLM [10] (Zhao et al., 2024), into our codebase. Each training batch includes samples from all training domains, with a batch size of 128 (reduced to 32 for Noise-Aware Generalization-Fashion). For relatively small datasets VLCS (Fang et al., 2013) and CHAMMI-CP (Chen et al., 2024b), experiments are run on a single NVIDIA RTX A6000 (48GB RAM) and three Intel(R) Xeon(R) Gold 6226R CPU @ 2.90GHz for 5000 steps. For Noise-Aware Generalization-Fashion (Xiao et al., 2015; 2017) and DomainNet-SN, experiments are run on four NVIDIA RTX A6000 (48GB RAM) and twelve Intel(R) Xeon(R) Gold 6226R CPU @ 2.90GHz for 15000 steps.

To determine the optimal learning rate, we sweep over some values in a range of values from $10^{-6}$ to $10^{-3}$ on a logarithmic scale. See Table 5 for the lr of specific methods on certain datasets.

# D    DETAILED RESULTS

Table 6, 7, 8 show the results for each domain in VLCS (Fang et al., 2013) and CHAMMI-CP (Chen et al., 2024b) datasets. For Table 6 and 7, the "Domain" column indicates the domain left out for testing. The OOD results show performance on this specific test domain, while the ID results reflect performance on the remaining training domains. For example, ID results for Caltech101 (Fei-Fei et al., 2004) indicate validation performance on a mixed dataset including LabelMe (Russell et al., 2008), VOC2007 (Everingham et al., 2010), and SUN09 (Choi et al., 2010). For CHAMMI-CP (Chen

---

[7]https://github.com/facebookresearch/DomainBed

[8]https://github.com/alexrame/fishr

[9]https://github.com/MediaBrain-SJTU/DISAM

[10]https://github.com/RyanZhaoIc/PLM/tree/main

et al., 2024b) results in Table 8, task1 shows the performance of ID task and task2 and task3 are both for OOD tasks. The OOD-AVG in the last column refers to the average performance across task2 and task3.

Looking at Table 6 and 7, we observe that DG methods generally perform better on ID tasks compared to LNL methods. However, the combination of LNL+DG methods shows greater promise in OOD tasks.

Table 6: **VLCS (Fang et al., 2013) OOD results** on Caltech101 (Fei-Fei et al., 2004), LabelMe (Russell et al., 2008), VOC2007 (Everingham et al., 2010) and SUN09 (Choi et al., 2010). Six groups of methods are presented: baseline (*ERM (Gulrajani & Lopez-Paz, 2020)*), DG methods (*SWAD (Cha et al., 2021), MIRO (Cha et al., 2022), ERM++ (Teterwak et al., 2023), SAGM (Wang et al., 2023)*), Robust-OOD methods (*VREx (Krueger et al., 2021), Fishr (Rame et al., 2022)*), Domain-aware optimization method (*DISAM (Zhang et al., 2024)*), LNL methods (*ELR (Liu et al., 2020), UNI-CON (Karim et al., 2022), DISC (Li et al., 2023), PLM (Zhao et al., 2024)*), and LNL+DG combination methods.

| Method | Caltech101 | LabelMe | SUN09 | VOC2007 | AVG |
|---|---|---|---|---|---|
| ERM | 97.73 | 64.36 | 73.47 | 72.84 | 77.10 |
| ERM++ | 98.45 | 63.78 | 72.06 | 76.42 | 77.68 |
| MIRO | 98.23 | 63.20 | 71.59 | 75.19 | 77.06 |
| SWAD | 99.29 | 62.12 | 74.37 | 80.49 | 79.07 |
| MIRO+SWAD | 98.76 | 61.79 | 73.84 | 77.05 | 77.86 |
| SAGM | 97.88 | 66.73 | 72.77 | 77.60 | 78.75 |
| SAGM+SWAD | 98.85 | 64.19 | 74.45 | 80.16 | 79.41 |
| Fishr | 97.17 | **67.61** | 66.31 | 72.30 | 75.85 |
| VREx | 96.82 | 63.65 | 69.66 | 73.93 | 76.02 |
| DISAM | 97.74 | 65.62 | 71.18 | 74.38 | 77.23 |
| ELR | 97.26 | 61.13 | 69.30 | 76.97 | 76.16 |
| DISC | 96.76 | 65.36 | 69.83 | 74.66 | 76.65 |
| UNICON | 99.51 | 61.37 | 73.46 | 75.21 | 77.39 |
| PLM | 97.17 | 64.83 | 72.73 | 67.65 | 75.60 |
| ERM++ + ELR | 98.23 | 62.54 | 74.13 | 77.55 | 78.11 |
| MIRO+UNICON | 99.01 | 61.29 | 71.88 | 72.66 | 76.21 |
| MIRO+SWAD+UNICON | **99.72** | 57.34 | 73.80 | 76.06 | 76.73 |
| MIRO+ELR | 98.23 | 62.59 | 69.95 | 79.27 | 77.51 |
| SWAD+ELR | 99.29 | 63.39 | **75.97** | 81.38 | **80.01** |
| MIRO+SWAD+ELR | 98.94 | 62.59 | 75.82 | **82.08** | 79.86 |

Table 7: **VLCS (Fang et al., 2013) ID results** on Caltech101 (Fei-Fei et al., 2004), LabelMe (Russell et al., 2008), VOC2007 (Everingham et al., 2010) and SUN09 (Choi et al., 2010). Six groups of methods are presented: baseline (*ERM (Gulrajani & Lopez-Paz, 2020)*), DG methods (*SWAD (Cha et al., 2021), MIRO (Cha et al., 2022), ERM++ (Teterwak et al., 2023), SAGM (Wang et al., 2023)*), Robust-OOD methods (*VREx (Krueger et al., 2021), Fishr (Rame et al., 2022)*), Domain-aware optimization method (*DISAM (Zhang et al., 2024)*), LNL methods (*ELR (Liu et al., 2020), UNICON (Karim et al., 2022), DISC (Li et al., 2023), PLM (Zhao et al., 2024)*), and LNL+DG combination methods.

| Method | Caltech101 | LabelMe | VOC2007 | SUN09 | AVG |
|---|---|---|---|---|---|
| ERM | 80.59 | 86.70 | 85.18 | 83.41 | 83.97 |
| ERM++ | 75.41 | 78.59 | 84.34 | 78.26 | 79.15 |
| MIRO | 80.63 | 89.70 | 86.82 | 86.70 | 85.96 |
| SWAD | **82.35** | **90.11** | 88.17 | 87.08 | **86.93** |
| MIRO+SWAD | 81.37 | 89.96 | **89.13** | 86.86 | 86.83 |
| SAGM | 81.89 | 90.10 | 88.59 | 86.56 | 86.78 |
| SAGM+SWAD | 81.84 | 89.80 | 88.73 | 86.16 | 86.63 |
| Fishr | 80.25 | 87.47 | 85.94 | 84.32 | 84.50 |
| VREx | 78.82 | 87.14 | 85.99 | 82.64 | 83.65 |
| DISAM | 80.04 | 86.24 | 86.21 | 85.11 | 84.40 |
| ELR | 82.59 | 88.70 | 87.14 | 86.81 | 86.31 |
| DISC | 80.93 | 85.94 | 84.08 | 84.20 | 83.79 |
| UNICON | 81.28 | 84.58 | 88.85 | 84.70 | 84.85 |
| PLM | 82.22 | 87.17 | 76.80 | 85.22 | 82.85 |
| ERM++ + ELR | 80.77 | 88.12 | 85.75 | 81.25 | 84.83 |
| MIRO+UNICON | 82.02 | 86.20 | 86.25 | 85.33 | 84.95 |
| MIRO+SWAD+UNICON | 81.44 | 86.08 | 85.95 | 81.82 | 83.82 |
| MIRO+ELR | 77.31 | 88.86 | 87.86 | 86.13 | 85.04 |
| SWAD+ELR | 81.75 | 89.90 | 88.16 | **87.55** | 86.84 |
| MIRO+SWAD+ELR | 81.97 | **90.11** | 88.22 | 86.80 | 86.78 |

Table 8: **CHAMMI-CP (Chen et al., 2024b) results**. Six groups of methods are presented: baseline (*ERM (Gulrajani & Lopez-Paz, 2020)*), DG methods (*SWAD (Cha et al., 2021), MIRO (Cha et al., 2022), ERM++ (Teterwak et al., 2023), SAGM (Wang et al., 2023)*), Robust-OOD methods (*VREx (Krueger et al., 2021), Fishr (Rame et al., 2022)*), Domain-aware optimization method (*DISAM (Zhang et al., 2024)*), LNL methods (*ELR (Liu et al., 2020), UNICON (Karim et al., 2022), DISC (Li et al., 2023), PLM (Zhao et al., 2024)*), and LNL+DG combination methods.

| Method | Task1(ID) | Task2(OOD) | Task3(OOD) | OOD-AVG |
|---|---|---|---|---|
| ERM | 79.22 | 56.80 | 25.35 | 41.08 |
| ERM++ | 72.49 | **62.45** | 26.64 | 44.55 |
| MIRO | 65.47 | 61.89 | **31.21** | **46.55** |
| SWAD | 73.91 | 61.99 | 25.32 | 43.66 |
| MIRO+SWAD | 67.31 | 62.24 | 29.40 | 45.82 |
| SAGM | 77.11 | 58.65 | 23.73 | 41.19 |
| SAGM+SWAD | 78.27 | 60.86 | 22.05 | 41.45 |
| Fishr | 67.65 | 57.12 | 27.24 | 42.18 |
| VREx | 67.30 | 58.28 | 25.87 | 42.08 |
| DISAM | 72.36 | 63.25 | 26.40 | 44.83 |
| ELR | 82.63 | 61.20 | 26.06 | 43.63 |
| DISC | 43.28 | 57.55 | 25.01 | 41.28 |
| UNICON | 76.72 | 58.57 | 25.46 | 42.02 |
| PLM | 70.77 | 62.57 | 26.31 | 44.44 |
| ERM++ + ELR | 75.72 | 59.11 | 24.96 | 42.04 |
| MIRO+UNICON | **84.52** | 61.71 | 25.17 | 43.44 |
| MIRO+SWAD+UNICON | 76.17 | 62.01 | 29.29 | 45.65 |
| MIRO+ELR | 74.54 | 58.90 | 23.65 | 41.28 |
| SWAD+ELR | 73.95 | 62.15 | 27.16 | 44.66 |
| MIRO+SWAD+ELR | 70.73 | 59.90 | 29.73 | 44.82 |

