# OpenReview forum: "In-N-Out: Robustness to In-Domain Noise and Out-of-Domain Generalization"
_ICLR.cc/2025/Conference — Submitted to ICLR 2025_

### Official Review · Reviewer_YbmP · 2024-10-17

**Soundness:** 2
**Presentation:** 3
**Contribution:** 1
**Rating:** 1
**Confidence:** 5

**Summary:**

This paper suggest to integrate noisy label problem and domain generalization problem together.

**Strengths:**

- This paper tried to integrate two well known problem setttings together since both problems should be managed together for real world settings.
- The authors may have studied previous researches thoroughly

**Weaknesses:**

- What is the novelty of the method proposed in this paper? What is different from naively combining all objective function from each area?
- There is a previous study which adjusted SAM being adequate for Domain generalization setting [1] and it says their method suggests a sharpness aware optimization for DG. Then, why $\mathcal{R}_{SAM}$ should be used?
- Too many hyperparameters. How can I balance $\alpha$. $\beta$, $\lambda$ and $\gamma$?

[1] Shin, S., Bae, H., Na, B., Kim, Y. Y., & Moon, I. C. (2024). Unknown Domain Inconsistency Minimization for Domain Generalization. arXiv preprint arXiv:2403.07329.

**Questions:**

- What is domain for Clothing1M dataset? In previous researches, I didn't see pointing out the domain shift issue of Clothing1M dataset. Also, why the number of image for Clothing1M is 100,000?

---

> ### Author Response · Authors · 2024-11-23
> **Responses to Weaknesses and Questions**
>
> Thank you for taking the time to review our work and for your thoughtful feedback. We also appreciate your recognition of the importance of the task in the world. Below are our responses.
>
> >W1. What is the novelty of the method proposed in this paper? What is different from naively combining all objective function from each area?
>
> We want to emphasize that novelty does not in proposing a “combination” method. Instead, those combination methods are just a way to explore the **new task** that jointly considers in-domain performance and out-of-distribution (OOD) generalization when data exhibits both in-domain noise and distribution shift. To the best of our knowledge, this task has not been investigated in any prior work.
>
> If this new task could be effectively solved by naively combining all objective functions from each area—there would be little justification for further research in this area. However, our findings reveal that the combination of noisy labels and distribution shifts introduces **unique challenges** that cannot be addressed by existing methods alone. This highlights the necessity of directly exploring this problem.
>
> Some key insights from our work include:
>
> 1, Multi-distribution settings can negatively impact noise detection methods.
> 2. Sample selection methods, while effective for handling label noise, can inadvertently distort distribution balance.
> 3. In noisy, multi-distribution data, data cleanliness plays a more critical role than domain balance in achieving robust performance.
>
> We believe our work underscores the significance of addressing this combined problem and opens up new avenues for further research, emphasizing the need for greater attention to this critical topic.
>
>
> >W2. There is a previous study which adjusted SAM being adequate for Domain generalization setting [1] and it says their method suggests a sharpness aware optimization for DG. Then, why R_{SAM} should be used?
>
> Thank you for sharing this reference. Here are a few clarifications:
>
> 1. Eq. 3 is not a specific loss function, but rather a unified general framework for combining different methods. Each term (e.g. L_{LNL}, L_{DG} ) represents a distinct loss defined by its respective method. We’ll update R_{SAM} as R_{DG} to present the general term of DG regularization.
>
> 2. We are eager to try the suggested method in the future; however, the GitHub link provided in the paper (https://github.com/SJShin-AI/UDIM) is currently unavailable and we can not get fair results within the rebuttal period.
>
> >W3. Too many hyperparameters. How can I balance \alpha, \beta, \lambda and \gamma?
>
> Balancing the weights of each loss term (\alpha, \beta, \lambda and \gamma) can be done using any validation set for a task.  In addition, most state-of-the-art methods balance similar, and often more, numbers of hyperparameters.   That said, our goal is not necessarily to say that this is the approach one should take.  Instead, as our task can be thought of as a combination of separately studied tasks, we explored an approach that simply combined methods from prior work on each individual task.  As discussed for W1, our analysis found that these methods often do not perform well, and that new challenges arise from the combination that did not come up when each task was explored alone.  Thus, this demonstrates that direct investigation of the combined task is required to improve performance.
>
> >Q1. What is domain for Clothing1M dataset? In previous researches, I didn't see pointing out the domain shift issue of Clothing1M dataset. Also, why the number of image for Clothing1M is 100,000?
>
> Clothing1M was collected from three different e-commerce websites (Ebay, Amazon, and Taobao), which can be thought of as distinct image domains as illustrated in Fig. 2, where images of the same "dress" class vary significantly—some show models with backgrounds, others have blank backgrounds, and some contain multiple objects.
>
> We agree that you are correct that the multi-distribution input issue has been overlooked in prior research on this dataset. As such, this is exactly where our work adds value—previous studies on LNL and DG often treat the tasks separately and assume ideal conditions for the other problem. For example, Clothing1M is widely used in LNL studies, where the focus has typically been on solving label noise, without addressing the domain shift issue. In contrast, our work aims to reflect more realistic application scenarios, where both noisy labels and domain shifts are present.
>
> As for the number of images, 100,000 is a typo. Thank you for pointing that out—it should be 1,000,000 images. It has been updated in the pdf.

---

> > ### Author Response · Authors · 2024-12-02
> >
> > Hello Reviewer YbmP- As we are coming to the last roughly 24 hours or so in the discussion period we were hoping you would review our rebuttal and consider improving your score.  Thank you for your efforts!

---

### Official Review · Reviewer_XACR · 2024-11-03

**Soundness:** 2
**Presentation:** 2
**Contribution:** 2
**Rating:** 5
**Confidence:** 4

**Summary:**

The paper proposes the In-N-Out task, which investigates the question of how to balance both in-distribution performance and out-of-distribution performance under the presence of label noise. Focusing on empirical analysis, the paper explores the possibility of combining loss functions/regularizers from learning with noisy labels (LNL) and domain generalization (DG), and demonstrates that (1) the performance of different combinations varies across datasets, and no single approach conclusively outperforms the rest, and (2) LNL approaches that are suitable for in-domain learning may not be suitable for out-of-domain generalization.

**Strengths:**

1. The paper aims to study a new task, In-N-Out, which focuses on the noise robustness for both in-domain and out-of-domain generalization, by examining which combination of different types of loss functions (such as LNL and DG losses) work the best.
2. The paper demonstrates that the ranking of different LNL algorithms can change when the empirical setup changes from one domain to multiple domains.
3. In-depth study on how do ELR, MIRO, UNICON algorithms behave in out-of-distribution generalization settings are performed. In particular, it empirically shows that the sampling strategy of UNICON may skew the domain distribution, hurting the generalization.

**Weaknesses:**

1. The paper aims to study what combination of different types of loss functions (DG, LNL, etc.) improves both ID and OOD performance under label noise. Although there are many choices for each type of loss function, only MIRO/SWAD is chosen for the DG component, and ELR/UNICON is chosen for the LNL component, without sufficient justifications. If the goal of the paper is to benchmark a combination of the algorithms, there seems to be a lack of support since only a few candidates are studied.
2. Figure 3 in Section 3.3.1 shows the experimental results of the ID performance with an increasing number of domains, but the discussion talks about OOD samples are hard for UNICON. Performances on OOD domains need to be compared to draw such a conclusion.
3. In Figure 4b, an insufficient number of baselines are compared (consider the LNL options and regularization/SAM in Equation 3). In addition, although SWAD exhibits better robustness to MIRO, they need to be not mutually-exclusive approaches, as they can be used together (shown in Table 1).
4. The flow of the paper can be improved by better motivating the problem statement and organizing the research questions in the analysis sections. It is a bit difficult for the reader to identify the main points from the chunks of information.

**Questions:**

1. Across the four types of losses in Equation (3), which one is more important towards in-domain/out-of-domain generalization when there is label noise in the training set? What do the weights of the losses in Equation (3) look like after tuning the hyperparameters? Do they provide additional insights on which component is contributing the most to the performances?
2. Model/checkpoint selection is a very important part of domain generalization and hyperparameter tuning. What model selection strategy is used here?
3. The experimental design is a bit confusing in Section 3.3.3. My interpretation is that the label noise rate is fixed across different sampling ratios (if it’s not fixed then the experiment is not a controlled study). It is surprising to see the in-distribution performance degrade drastically as the number of samples increases. The arguments about “maintaining balance” and “increased noise” in line 407 do not sound reasonable. How does the sampling ratio affect these two characteristics? Please provide more justifications.
4. Minor typos: line 186 “textit”.

---

> ### Author Response · Authors · 2024-11-23
> **Responses to Weaknesses**
>
> Thank you for taking the time to review our work and for your thoughtful feedback. We also appreciate your recognition of the new task. Below are our responses.
>
> >W1. MIRO/SWAD is chosen for the DG component, and ELR/UNICON is chosen for the LNL component, without sufficient justifications.
>
> We selected MIRO/SWAD for the DG component and ELR/UNICON for the LNL component based on their strong performance in their respective groups on our tasks. Thus, given the large number of possible combinations, we focused on these specific pairs for in-depth analysis.
>
> Due to time constraints, we only completed tuning one additional combination, SAGM+UNICON, on the VLCS dataset. The results for this combination showed similar trends to those of other combinations, where the combined method performed worse than the individual methods. We will include the results for the full datasets in the camera-ready version.
>
> | Method        | ID     | OOD    | AVG   |
> |---------------|--------|--------|-------|
> | MIRO          | 85.96  | 77.06  | 81.51 |
> | SAGM          | 86.78  | 78.75  | 82.77 |
> | UNICON        | 84.85  | 77.39  | 81.12 |
> | MIRO+UNICON   | 84.95  | 76.21  | 80.58 |
> | SAGM+UNICON   | 84.13  | 76.89  | 80.51 |
>
> >W2. Figure 3 in Section 3.3.1 shows the experimental results of the ID performance with an increasing number of domains, but the discussion talks about OOD samples are hard for UNICON. Performances on OOD domains need to be compared to draw such a conclusion.
>
> We believe the confusion is from the term "OOD," which is not entirely accurate in this context. We updated the wording in the PDF for clarity. Here, "OOD" was intended to refer to samples from different domains. During sample selection across all domains, minority domains may be mistakenly treated as "OOD" samples. UNICON faces the challenge of distinguishing between noise and samples from other domains. This challenge is evident in Fig. 7, where domains with fewer samples are selected less frequently. For instance, in the “person” class, the representation of Caltech data decreases significantly from 25.87% to 11.92% in the selected samples.
>
> >W3. In Figure 4b, an insufficient number of baselines are compared (consider the LNL options and regularization/SAM in Equation 3). In addition, although SWAD exhibits better robustness to MIRO, they need to be not mutually-exclusive approaches, as they can be used together (shown in Table 1).
>
> Thanks for the suggestions! Figure 4 aims to highlight the impact of noise on DG methods, which is why we did not include other LNL methods in this comparison. We will update this plot by adding SWAD+MIRO, VREx, and Fishr these DG methods. However, due to the larger dataset and multiple runs at different noise ratios, the results are not ready yet. We will update the figure in the PDF as soon as possible.
>
> >W4. The flow of the paper can be improved by better motivating the problem statement and organizing the research questions in the analysis sections. It is a bit difficult for the reader to identify the main points from the chunks of information.
>
> Thank you for the suggestions! We’ve revised the structure of Section 3.3: Analysis in the updated manuscript to improve the flow and better highlight the research questions motivating our study. The section is now organized around the following key questions:
>
> 3.3.1 How Does Multi-Source Training Data Impact LNL Methods?
>
> 3.3.2 How Does Noisy Training Data Impact DG Methods?
>
> 3.3.3 Cleaner vs. Balanced: Which Is More Critical for ID and OOD Performance?
>
> 3.3.4 What Are the Insights for Combining LNL and DG Methods?
>
> * Exploring combinations of disjoint losses.
> * Training with domain labels.
>
> We hope this restructured presentation clarifies the key points of our analysis and makes the findings easier to follow. Let us know if further adjustments are needed!

---

> ### Author Response · Authors · 2024-11-23
> **Responses to Questions**
>
> >Q1. Across the four types of losses in Equation (3), which one is more important towards in-domain/out-of-domain generalization when there is label noise in the training set? What do the weights of the losses in Equation (3) look like after tuning the hyperparameters? Do they provide additional insights on which component is contributing the most to the performances?
>
> To clarify, Eq. (3) serves as a general framework for combining different losses in the context of DG and LNL methods. Each loss component (DG/LNL) can be decomposed into the terms outlined in Eq. (3). However, in practice, we did not individually tune the hyperparameters for each term. Instead, we employed two sets of losses and optimizers and trained with both optimizers simultaneously.  However, to provide some insight, we conducted experiments with the MIRO+UNICON combination on the CHAMMI-CP dataset. This has the expected result based on our other observations-  increasing the weight of MIRO improves OOD performance as DG methods like MIRO were designed to improve performance on those samples, while increasing the weight of UNICON enhances ID performance as LNL methods are evaluated on ID gains.  We will provide an expanded study in our camera ready.
>
> | Weight Ratio (MIRO:UNICON) | ID    | OOD   |
> |----------------------------|-------|-------|
> | 0.2                        | 85.50 | 42.82|
> | 0.5                        | 85.48 | 42.62 |
> | 1.0                        | 84.52 | 43.44 |
> | 1.2                        | 79.30 | 45.04 |
> | 1.5                        | 78.17 | 46.07 |
>
>
> >Q2. Model/checkpoint selection is a very important part of domain generalization and hyperparameter tuning. What model selection strategy is used here?
>
> The checkpoint is selected and saved by the in-domain validation dataset.
>
> >Q3. The experimental design is a bit confusing in Section 3.3.3. My interpretation is that the label noise rate is fixed across different sampling ratios (if it’s not fixed then the experiment is not a controlled study). It is surprising to see the in-distribution performance degrade drastically as the number of samples increases. The arguments about “maintaining balance” and “increased noise” in line 407 do not sound reasonable. How does the sampling ratio affect these two characteristics? Please provide more justifications.
>
> The noise rate is fixed for the input dataset, but the noise rate in the selected samples may vary.
>
> The selection ratio can be interpreted as an indicator of how confident we are about the cleanliness of the samples. When the selection ratio is low, only a small number of samples are selected, but they are highly confident to be clean. Conversely, a high selection ratio results in a larger number of selected samples, but with lower average confidence in their cleanliness. This trend is illustrated in Fig. 5 (b). For example, as the selection ratio increases, the percentage of clean samples in the SUN09 dataset decreases, which is visible in the ratio of dark green to light green regions.
>
> Additionally, with a higher selection ratio, the distribution distances between different domains tend to shrink. For instance, at a ratio of 0.2, the number of samples from VOC2007 is nearly double that of LabelMe, while at a ratio of 0.8, the number of images from these two domains becomes more comparable.
>
> Thus, the choice of selection ratio represents a trade-off between cleanliness and domain distribution balance. Our findings suggest that, for robustness enhancement, quality is more crucial than quantity.
>
> >Q4. Minor typos: line 186 “textit”.
>
> Thanks for pointing this out, and we’ve corrected this issue.

---

> > ### Comment · Reviewer_XACR · 2024-11-26
> >
> > Thank you for the detailed responses. I think most of my questions and concerns are addressed. The paper’s goal of conducting a comprehensive study on the intersection between DG and LNL is important and well-motivated. It also provides interesting findings for SWAD, ELR, and UNICON, and how practitioners should be careful about using these algorithms under DG settings (w/ or w/o LNL), though I believe providing more insights or empirical supports could strengthen the statements even more.
> >
> > My other concern still lies in the way the claims are presented and supported. I could find some insights by carefully reading the paper, but I also cannot be certain about these because they could be dataset-specific results. I would suggest the authors:
> >
> > 1. Before asking a question, can we motivate in some way that explains its importance?
> > 2. After asking a question such as “How does noisy training data impact DG methods?”, present the conclusion or statements immediately or in italic font or boxes to highlight, so that the messages are clear.
> > 3. For the questions in analysis 3.3, can these separate questions in parallel be structured? For example, why is it important to investigate “How do LNL methods respond to **multi-source** (is it multi-distribution? what’s its definition?) data?” After drawing the conclusions for this question, can it be connected with “How sensitive are DG methods to noise?”. These two seem very logically connected questions. It might be better to structure them as follows:
> >     - We begin by investigating the impact of domains on LNL tasks (3.3.1). We find that increasing the number of training domains complicates ID LNL. Then we ask ourselves: Can this be resolved by looking at DG approaches (3.3.2)? In addition, what is the corresponding OOD LNL in this case? We realize that training with SWAD is generally better than ERM, and the reason why UNICON may not perform well is because of the skew in domain distributions, and so on and so forth.
> > 4. Perhaps after exploring these directions, you can suggest a comprehensive recipe for choosing the LNL/DG method (e.g. Since SWAD consistently outperforms other DG baselines, it might be worthwhile to suggest using SWAD whenever possible and providing some deeper analysis).
> >
> > Besides, I think it’s better to only cite a method/dataset when it appears in the paper for the first time. It would make the paper much easier to read.

---

> > > ### Author Response · Authors · 2024-11-28
> > >
> > > Thank you for recognizing the motivation and significance of our work, as well as for providing detailed suggestions to refine our writing. Your feedback is valuable. Below, we summarize the updates we’ve made to the PDF in response to your suggestions:
> > >
> > > >1. Before asking a question, can we motivate in some way that explains its importance?
> > >
> > > The motivation of the first two questions is added at the beginning of 3.3 -- why NAG cannot be effectively solved using a single LNL or DG method alone.
> > >
> > > The motivation for the third question at 3.3.3 is also added:
> > >
> > > “As highlighted in earlier sections, imbalanced domain distributions pose additional challenges for LNL methods, while noise introduces difficulties for DG methods. This raises the question: how can we strike a balance between cleanliness and distributional balance?”
> > >
> > > >2. After asking a question such as “How does noisy training data impact DG methods?”, present the conclusion or statements immediately or in italic font or boxes to highlight, so that the messages are clear.
> > >
> > > We’ve adopted this approach in the revised PDF. Key conclusions are now highlighted immediately after the questions in italics:
> > >
> > > 3.3.1 How Does Multi-distribution Training Data Impact LNL Methods?
> > >
> > > *The performance of LNL methods declines when additional data sources with diverse distributions are introduced, with sample-selection methods being particularly impacted.*
> > >
> > > 3.3.2 How Does Noisy Training Data Impact DG Methods?
> > >
> > > *Performance across domains shows varying levels of decline under noisy conditions, highlighting the sensitivity of DG methods to noise. SWAD+MIRO demonstrates exceptional resilience to noise.*
> > >
> > > 3.3.3 Cleaner vs. Balanced: Which Enhances ID and OOD Performance?
> > >
> > > *Quality outweighs quantity in enhancing robustness.*
> > >
> > > 3.3.4 What Are the Insights for Combining LNL and DG Methods?
> > >
> > > *Exploring combinations of disjoint losses and utilizing domain labels are promising directions.*
> > >
> > > >3. For the questions in analysis 3.3, can these separate questions in parallel be structured?
> > >
> > > To improve the clarity and flow of Section 3.3, we’ve restructured the introduction to the analysis. The revised first paragraph now clearly sets up the motivation and progression of the questions, presenting them in a parallel structure.
> > >
> > > “Since NAG is a composite task that integrates two interrelated challenges, LNL and DG, our analysis begins by examining how introducing another factor affects the traditional task. Specifically, we investigate: How does multi-distribution data impact LNL methods? and How does noisy data impact DG methods? These questions address the core issue of \textbf{why NAG cannot be effectively solved using a single LNL or DG method alone}. Following this, we explore LNL and DG's interaction by examining the trade-off between prioritizing cleaner samples or maintaining balanced distributions. Finally, we conclude by offering insights and recommendations for addressing NAG effectively.”
> > >
> > > >4. Perhaps after exploring these directions, you can suggest a comprehensive recipe for choosing the LNL/DG method
> > >
> > > We added an additional subsection in *3.3.4 What Are the Insights for Combining LNL and DG Methods?* where we provide a discussion of our cross-dataset observations:
> > >
> > > **Regularization-based techniques are more effective.** Table 1 shows an interesting pattern: datasets where domain shifts are more significant (VLCS and NAG-Fashion) regularization-based methods from the DG literature are generally more effective, whereas on CHAMMI-CP where label noise is more of an issue, LNL regularization is more effective (e.g., ELR). Combining these generally improves performance. Other LNL methods that try to correct labels, e.g., UNICON, can be effective in the low domain shift setting when combined with regularization techniques Table 1 or when domain labels are available to minimize domain shift Table 2, discussed below.
> > >
> > >
> > > In our conclusion, we also add the follow recommendations:
> > >
> > > We provide the following specific recommendations based on our experiments. Generally there
> > > are two components to any dataset: the amount of noise and the strength of the domain shift. The
> > > inherent inability to separate these two factors mean that regularization-based techniques (e.g., SWAD
> > > and MIRO) are more effective. From here, our recommendations diverge based on the amount of
> > > domain shift present. In cases of high domain shift (e.g., VLCS and NAG-Fashion), domain labels
> > > are required to use other techniques (e.g., pseudo-labeling UNICON for addressing noisy labels), as
> > > they the effect of domain shifts are minimized. If the domains during training are smaller than those
> > > seen at test time (e.g., NAG-Fashion), then additional regularization may be required. In cases of low
> > > domain shift (e.g., CHAMMI-CP) combining regularization with other techniques like UNICON can
> > > be used immediately even in cases of high noise, but too much regularization can be detrimental.

---

> ### Author Response · Authors · 2024-11-28
>
> >Besides, I think it’s better to only cite a method/dataset when it appears in the paper for the first time. It would make the paper much easier to read.
>
> Thank you for the suggestion. We have removed some of redundant citations to better balance readability and clarity in our paper.
>
> Note that we provided an updated pdf, and will continue to edit and refine our paper for any camera ready.

---

> > ### Author Response · Authors · 2024-12-02
> >
> > Hello Reviewer XACR- As we are counting down to the last roughly 24 hours we were wondering if there are any questions or if you have any response to our rebuttal and subsequent discussion? We would be happy to discuss this further to help improve your score or provide concrete justification as to what needs to be improved. Thank you!

---

### Official Review · Reviewer_eScZ · 2024-11-05

**Soundness:** 3
**Presentation:** 2
**Contribution:** 2
**Rating:** 5
**Confidence:** 4

**Summary:**

This paper studies the combination of Learning with Noisy Labels (LNL) and Domain Generalization (DG), introducing a task called In-N-Out that aims to simultaneously handle label noise and domain shifts. The authors evaluate various LNL and DG methods and their combinations on three real-world datasets and one synthetic dataset, providing empirical analysis of how these methods interact.

**Strengths:**

1. The paper provides a thorough empirical study comparing multiple methods and their combinations. The experimental analysis examines different aspects like multi-source impact on LNL methods, noise sensitivity of DG methods, and trade-offs between data cleanliness and domain balance.

2. The authors provide some interesting observations through their experiments including method combinations sometimes performing worse than individual approaches. These observations, while not groundbreaking, provide useful insights.

**Weaknesses:**

1. I feel the fundamental contributions are limited. The paper combines two existing problems (LNL and DG) without proposing novel technical solutions. The framework is a straightforward combination of existing loss functions and methods.

2. Another major weakness is that the "task" being proposed seems artificial rather than motivated by real application needs. While real data may contain both noise and domain shifts, treating these as a single unified problem rather than addressing them separately needs stronger motivation. The paper doesn't clearly demonstrate why existing approaches of handling these issues separately is insufficient.

**Questions:**

Please check the weakness

---

> ### Author Response · Authors · 2024-11-23
> **Responses to Weaknesses**
>
> Thank you for taking the time to review our work and for your thoughtful feedback. We also appreciate your recognition of our study. Below are our responses.
>
> >W1.I feel the fundamental contributions are limited. The paper combines two existing problems (LNL and DG) without proposing novel technical solutions. The framework is a straightforward combination of existing loss functions and methods.
>
> We respectfully disagree with the interpretation of our work as merely a straightforward framework combining existing methods. Our key contribution lies in identifying the gaps and challenges that arise when integrating these approaches. If this problem could be effectively addressed by simply applying existing LNL or DG methods—or their combination—there would be little need for further research in this area.
>
> However, our findings show that the combination of noisy labels and distribution shifts introduces unique challenges that cannot be addressed by existing methods alone. Notably, no single DG or LNL method consistently achieves the best performance across all benchmarks, and combining methods does not always yield superior results. For instance, in the fashion dataset experiments, the DG method MIRO+SWAD achieves the highest average performance (ID: 91.02, OOD: 60.87), while the LNL method UNICON performs best individually (ID: 87.31, OOD: 53.85). However, their combination, MIRO+SWAD+UNICON (ID: 86.09, OOD: 57.18), falls short of the performance achieved by MIRO+SWAD+ELR (ID: 91.48, OOD: 63.53). This highlights the necessity of directly exploring this problem.
>
> Some key insights from our work include:
>
> 1. Multi-distribution settings can negatively impact noise detection methods.
>
> 2. Sample selection methods, while effective for handling label noise, can inadvertently distort distribution balance.
>
> 3. In noisy, multi-distribution data, data cleanliness plays a more critical role than domain balance in achieving robust performance.
>
> We believe our work underscores the significance of addressing this combined problem and opens up new avenues for further research, emphasizing the need for greater attention to this critical topic.
>
> >W2. Another major weakness is that the "task" being proposed seems artificial rather than motivated by real application needs. While real data may contain both noise and domain shifts, treating these as a single unified problem rather than addressing them separately needs stronger motivation. The paper doesn't clearly demonstrate why existing approaches of handling these issues separately is insufficient.
>
> We respectfully disagree and emphasize that the proposed "task" is directly driven by real-world applications. The datasets used in this paper—VLCS, Clothing1M, and CHAMMI-CP—all contain **real** label noise and input distribution shifts, highlighting the practical relevance of this problem.
>
> For instance, consider CHAMMI-CP:
>
> * Label noise arises from weak treatment effects, where cells that resemble the control group (no drug) are mislabeled due to subtle treatment differences.
>
> * Input distribution shift stems from experimental technical variations. Cells treated with the same drug can exhibit varying visual effects depending on factors like well/plate conditions.
>
> Methods with the best average performance in each group are selected and shown in the table below (more details can be found in Table 1 of the main paper). Existing LNL and DG methods individually fail to address this combined challenge effectively. In many cases, their performance on both in-domain (ID) and out-of-domain (OOD) benchmarks is even worse than the baseline ERM method. By contrast, the combination method demonstrates promising results, though challenges remain. Notably, combining the best individual DG method (MIRO) with the best individual  LNL method (ELR) does not yield the best performance, highlighting the complexities of addressing the combined problem.
>
> | Method      |    ID |   OOD |    AVG |
> |:------------|------:|------:|-------:|
> | ERM         | 79.22 | 41.08 | 60.15  |
> | MIRO        | 65.47 | 46.55 | 56.01  |
> | VREx        | 74.78 | 44.81 | 59.80  |
> | DISAM       | 72.36 | 44.83 | 58.60  |
> | ELR         | 82.63 | 43.63 | 63.13  |
> | MIRO+ELR | 74.54 | 41.28 | 57.91 |
> | MIRO+UNICON | 84.52 | 43.44 | **63.98**  |
>
>
> The same trends are observed across other datasets. No single LNL or DG method achieves consistent ID and OOD performance across all benchmarks, and even the combination of state-of-the-art methods struggles with this problem. This underscores not only the insufficiency of existing approaches but also the limitations of current combination methods, highlighting the critical need for further investigation.

---

> > ### Author Response · Authors · 2024-12-02
> >
> > Hello Reviewer eScZ- As we are coming to the last roughly 24 hours or so in the discussion period we were hoping you would review our rebuttal and consider improving your score.  Thank you for your efforts!

---

### Official Review · Reviewer_yqHM · 2024-11-07

**Soundness:** 3
**Presentation:** 3
**Contribution:** 2
**Rating:** 5
**Confidence:** 4

**Summary:**

The paper does an empirical comparison of methods for domain generalization and for handling label noise, under a suite of vision domain generalization datasets that either contain label noise to begin with, or where label noise is added synthetically.

With a large scale comparison of methods for labels noise, domain generalization, and their combination, the authors report several findings. Among these findings are the statements that no single method constantly performs better than the others across all benchmarks, and that methods that handle label noise using sample selection tend to underperform when there is data from several domains.

**Strengths:**

The paper touches upon the combination of two important problems, label noise and domain generalization. It makes a very thorough empirical comparison, and the experiments seem well executed. The observation about selection methods for label noise being less effective with data from diverse domains seems interesting.

**Weaknesses:**

My main concern about the paper is that the main novelty is in setting up and running the large scale comparison. While some of the conclusions from the experiments are novel insights, I am not sure they are significant enough to warrant acceptance.
Some novelty either in terms of methods, or data selection algorithms, or experiments that sugges a way forward might have contributed to novelty.

Beyond the concern about novelty, I think there are a few other points worth mentioning.
* The choice of the name In-n-Out for the setting should be revised. There is an OOD-generalization method from an ICLR 2021 paper with the same name [1].
* One of the main findings is that no single method consistently outperformed others. The authors also state that this is surprising (line 531), however I do not understand why this is a surprise. In OOD-Generalization it is unclear whether there is one method that outperforms the other across all "natural" distribution shifts, and this is reflected in several works in the literature. For instance by [2] for spurious correlations benchmarks, or the WILDS dataset leaderboard for other benchmarks.
* Beyond not having a clear winner (which is reasonable and expectable), it is unclear what in the dataset makes one method more suitable for it than the other. Hence it is not really clear which of the empirical conclusions lead to practical advice for researchers and engineers. Besides the insight about the sample selection methods, the rest of the conclusions don't give a clue for when should one prefer ERM, one DG method or another, and one label noise method vs. the other.


[1] Xie, Sang Michael, et al. "In-N-Out: Pre-Training and Self-Training using Auxiliary Information for Out-of-Distribution Robustness." International Conference on Learning Representations 2021.‏
[2] Bell, Samuel J., Diane Bouchacourt, and Levent Sagun. "Reassessing the Validity of Spurious Correlations Benchmarks." arXiv preprint arXiv:2409.04188 (2024).‏

**Questions:**

* The paper distinguishes Out-of-Domain methods and OOD Robustness methods. That categorization is unclear to me, what is the definition that discerns of these categories? For instance, why does Fishr or IRM belong into one category and not the other?
* Line 52 states that "some DG methods show implicit OOD-robustness under noise", referring to GroupDRO, Fishr and others. Does this statement rely on an empirical or theoretical observation? Where in the papers on these methods is this claim shown?
* Line 186 has a missing \ before "textit"

---

> ### Author Response · Authors · 2024-11-23
> **Responses to Weaknesses**
>
> Thank you for taking the time to review our work and for your thoughtful feedback. We also appreciate your interest in this topic. Below are our responses.
>
> >W1. My main concern about the paper is that the main novelty is in setting up and running the large scale comparison.
>
> We'll explain the novelty of the work from the following three aspects: what, why, and how.
>
> **What**
>
> We introduce a new research **task** that considers in-domain performance and out-of-distribution (OOD) generalization when data exhibits both in-domain noise and distribution shift.
>
> **Why**
>
> 1. Real-world applications: This problem is highly motivated by challenges in real-world datasets, such as:
>
> Clothing1M: An online multi-source dataset with label noise.
>
> CHAMMI-CP: A biomedical cell image dataset with technical variations.
>
> 2. Research gap: To the best of our knowledge, no prior work has tackled the combined challenge of noisy labels and distribution shifts simultaneously.
>
> 3. Methodological limitations: Neither state-of-the-art (SoTA) methods for learning with noisy labels nor domain generalization individually—or in combination—fully solve this problem, see Table 1 for detailed results.
>
> **How**
>
> We provide valuable insights into this new problem by highlighting challenges and proposing solutions:
>
> 1. Multi-distribution settings can negatively affect noise detection methods.
> 2. Sample selection methods, while addressing label noise, can unintentionally skew distribution balance.
> 3. In noisy multi-distribution data, the cleanliness of the data plays a more critical role than the domain balance in achieving robust performance.
> 4. Utilizing disjoint loss and domain labels can improve performance.
>
> **Our key message**
>
> While we present a combined method ("A+B") that performs better than standalone solutions ("A" or "B") for this problem, the main contribution lies not just in the combination itself, but in **identifying the gaps and challenges** that arise even when combining existing approaches.
>
> We believe this opens up avenues for further exploration and emphasizes the need for more research attention on this critical topic.
>
>
> >W2. The choice of the name In-n-Out for the setting should be revised.
>
> Thanks for the advice! We updated the name in the manuscript as Noise-Aware Generalization (NAG).
>
> >W3. One of the main findings is that no single method consistently outperformed others. The authors also state that this is surprising (line 531), however I do not understand why this is a surprise.
>
> We thank the reviewer for sharing this reference. We agree with the observation that “benchmarks often disagree, and methods that perform well on one benchmark perform poorly on others.”
>
> The reason we describe this as a "surprise" is that we initially expected the **combination problem (jointly addressing noisy labels and distribution shifts) to be solvable using a straightforward combination method (e.g., state-of-the-art LNL + state-of-the-art DG)**. However, our findings demonstrate that this is **not** the case.
>
> To put it another way: if this problem could be effectively addressed by simply applying existing LNL or DG methods—or their combination—then there would be little value in further investigating this topic. This surprising result instead underscores the necessity of directly exploring this problem, as the combination of noisy labels and distribution shifts presents unique challenges that cannot be resolved by combining existing state-of-the-art approaches.
>
> >W4.Beyond not having a clear winner (which is reasonable and expectable), it is unclear what in the dataset makes one method more suitable for it than the other. Hence it is not really clear which of the empirical conclusions lead to practical advice for researchers and engineers. Besides the insight about the sample selection methods, the rest of the conclusions don't give a clue for when should one prefer ERM, one DG method or another, and one label noise method vs. the other.
>
> Thank you for raising this point and highlighting the importance of providing practical guidance. To clarify, the primary aim of our work is not to rank existing methods but to provide insights that can inform the design of future methods for tackling the joint challenges of noisy and multi-distribution data. Below, we summarize the key practical advice from three perspectives:
>
> 1. How does multi-distribution data impact LNL methods?
>
> LNL methods suffer from distinguishing between noise and samples from different distributions. (Sec. 3.3.1)
>
> 2. How do noise and LNL methods impact DG methods?
>
> Sample selection can skew the original domain distribution, potentially degrading OOD performance. (Sec. 3.3.2)
>
> 3. How do we balance the combination between LNL and DG?
>
> Prioritize quality over quantity for robust performance. (Sec. 3.3.3)
>
> Combining with disjoint loss and utilizing domain labels can be effective. (Sec. 3.3.4)

---

> ### Author Response · Authors · 2024-11-23
> **Responses to Questions**
>
> >Q1: The paper distinguishes Out-of-Domain methods and OOD Robustness methods. That categorization is unclear to me, what is the definition that discerns of these categories? For instance, why does Fishr or IRM belong into one category and not the other?
>
> As we note in the introduction of our paper, there is some DG work that explores the effect of label noise on DG methods, such as [1].  In this paper, the authors show some methods like Fishr and IRM are more robust to label noise than others. As we will discuss in the response to the next question, this is based on a number of observations made in these papers.  Thus, one can think of these methods as those that were already shown to have some robust properties to label noise, although many do not go so far as to incorporate mechanisms to address label noise as we do in our paper.
>
> >Q2: Line 52 states that "some DG methods show implicit OOD-robustness under noise", referring to GroupDRO, Fishr and others. Does this statement rely on an empirical or theoretical observation? Where in the papers on these methods is this claim shown?
>
> That statement in Line 52 is also based on findings from papers like [1] (which we cite on that line as evidence) that explore the effect of label noise on DG methods, which provide both theoretical and empirical evidence for OOD-robustness for methods such as such as GroupDRO and Fishr.  I.e., this is not something shown in the papers that introduced those methods themselves, but rather on follow-up work analyzing those methods.
>
> [1] Rui Qiao and Bryan Kian Hsiang Low. Understanding domain generalization: A noise robustness perspective. ICLR, 2024.
>
> >Q3. ​​Line 186 has a missing \ before "textit".
>
> Thanks for pointing out, and we’ve corrected it in the manuscript.

---

> > ### Comment · Reviewer_yqHM · 2024-11-27
> >
> > Thank you for the response, it clarifies some points well and I appreciate the reference to the paper on noise robustness.
> >
> > Personally, I still find the novelty in the paper insufficient for a venue like ICLR. It is interesting to explore the combination of domain shifts and label shifts, but I think a publication on this requires further insights such as a new method, or mathematical understanding of the mechanism behind the observed empirical behaviors.
> >
> > To further comment on the response to question 1 that I asked: in the context of the paper, I see why it makes sense to divide the OOD robustness methods into two groups where one is robust to label noise and the other isn't. But then the difference between the names "Out-of-Domain methods" and "OOD Robustness methods" does not reflect the core different between them, which the authors claim is their robustness to label noise.
> >
> > Thanks again for the detailed response.

---

> > > ### Author Response · Authors · 2024-11-28
> > >
> > > >To further comment on the response to question 1 that I asked: in the context of the paper, I see why it makes sense to divide the OOD robustness methods into two groups where one is robust to label noise and the other isn't. But then the difference between the names "Out-of-Domain methods" and "OOD Robustness methods" does not reflect the core different between them, which the authors claim is their robustness to label noise.
> > >
> > > Thank you for your comment.  As you say, it makes sense in the context of our paper, which is how they are different from other methods in the ways that one can argue matters most in context.  However, we are happy to change this if the reviewer has a specific suggestion on how this can be improved.

---

> > > > ### Comment · Reviewer_yqHM · 2024-11-28
> > > >
> > > > Thank you for the response. My small suggestion was to have the difference in the names reflect the actual difference between the methods. If the difference between the methods is their robustness to label noise, then it seems like Domain-shift methods and Domain+Label-shift methods, or anything along that line makes more sense than the current naming. However, this is not a crucial point in my view, just a small suggestion to improve the paper. Thanks again.

---

> > > > > ### Author Response · Authors · 2024-12-02
> > > > >
> > > > > Hello Reviewer yqHM- As we are counting down to the last roughly 24 hours we were wondering if there are any questions or if you have any response to our rebuttal and subsequent discussion? We would be happy to discuss this further to help improve your score or provide concrete justification as to what needs to be improved. Thank you!

---

> ### Author Response · Authors · 2024-11-28
>
> >Personally, I still find the novelty in the paper insufficient for a venue like ICLR. It is interesting to explore the combination of domain shifts and label shifts, but I think a publication on this requires further insights such as a new method, or mathematical understanding of the mechanism behind the observed empirical behaviors.
>
> Thank you for the opportunity to discuss this further!  Our paper presents a new task that combines domain shifts and noise robustness.  Generally speaking, based on the evidence provided in prior iterations of ICLR and other ML venues, there are two basic requirements for these papers to be accepted for publication: demonstrating that the new task is important and then also showing that it has new challenges not present in prior work. It appears as though reviewers agree that the problem is important and this presents a new challenge not present in either DG and LNL alone.
>
> One specific issue is the problem that stems from trying to distinguish between domain shifts and noisy labels, resulting in regularization-based methods being more effective.  While we could provide some additional analysis on this specific point, there already exists supporting evidence that supports this from the out-of-domain detection literature [A], making a thorough analysis arguably perhaps redundant on this point.  Additionally, some prior work published at ICLR 2024 already provides a thorough analysis as to why domain generalization methods find noisy labels challenging [B], making an analysis as the reviewer suggested here redundant as this issue is known.  However, what [B] did not do and what our paper focuses on is what happens to mitigate the effect of label noise by integrating LNL literature, which has ignored the effect of the domain shift.  On this point, in Section 3.3 we did provide a through analysis and found that the challenge of discriminating between label noise and domain shifts means that some LNL techniques like sample selection means that easier domains/samples are retained, resulting in a skewed distribution being selected.  For example, in Figure 4 almost every car sample from LabelMe was selected by UNICON, and relatively few from Sun09 or VOC2007, resulting a biased training process that reduces performance.  We then further verify that these methods can boost performance if the domain shifts are controlled (Table 2).  Thus, it is clear that these methods are suffering from the challenge methods are faced when trying to discriminate between out-of-domain samples and label noise. We again highlight is well understood as a challenging problem to address from the OOD detection literature [A], making additional analysis redundant on this point.  However, where our task differs from OOD detection is that noisy label sources can be leveraged to better identify what samples might be noisy.  In other words,  for our task one can use the fact that cheetahs and leopards are more often confused for each other, thereby providing stronger  guidance when a sample might be a domain shift or label noise [C], which is not an assumption possible in general OOD detection and a clear avenue for future work.
>
> Finally, we note that while new methods are not typically required for providing a contribution when a new task is introduced, there are also countless papers that have been accepted (and likely the majority) that effectively combine two or more methods from prior work. Our paper provides more than just two combinations, and these are from different bodies of work that have not been previously considered to work in concert with each other.  As such, our contribution to the modeling aspect is arguably stronger than many of these papers.  In addition, other papers like [B] have also been published at ICLR without presenting "new" methods, focusing instead on an analysis.
>
> Given the discussion above, we would appreciate some specific guidance as to what you feel would make our paper stronger.  We acknowledge that there is always room for improvement, but without clear direction as to what exactly the reviewer feels is missing it makes it impossible to address your comments, as much of the clear additional analysis we could do is already present in prior work.  At the very least, we are happy to articulate these points so the contributions are clearer in a revised version.
>
> [A] A noisy elephant in the room: Is your out-of-distribution detector robust to label noise? CVPR 2024
>
> [B] Understanding Domain Generalization: A Noise Robustness Perspective. ICLR 2024
>
> [C] LNL+K: Enhancing Learning with Noisy Labels Through Noise Source Knowledge Integration. ECCV, 2024

---

> > ### Comment · Reviewer_yqHM · 2024-11-28
> >
> > Thank you for the response and further efforts. As your latest response says, the combination of label noise and spurious correlations has been mentioned in past work (e.g. reference [B] in your comment, which aims to contribute mathematical understanding of the phenomenon). Therefore, respectfully, I do not think this is an entirely new task (one might argue that it is under-explored, but that is a different matter). Hence I'd imagine that one meaningful contribution would be to suggest a technically innovative solution to this problem, providing mathematical understanding as to why method combinations fail, or proving that some cases simply cannot be solved without further assumptions. My main comment is that while I did learn some empirical fact from the paper, I did not gain any new understanding about the problem, or learn about a novel solution to it. The initial insight is interesting and poses a good motivation for making a contribution on the topic, but I do not think that this insight alone leads to better understanding. This is my personal point of view, and it is reasonable if the authors or other reviewers hold a different one. Thanks again for the discussion.

---

> ### Author Response · Authors · 2024-11-28
>
> >Thank you for the response and further efforts. As your latest response says, the combination of label noise and spurious correlations has been mentioned in past work (e.g. reference [B] in your comment, which aims to contribute mathematical understanding of the phenomenon). Therefore, respectfully, I do not think this is an entirely new task (one might argue that it is under-explored, but that is a different matter). Hence I'd imagine that one meaningful contribution would be to suggest a technically innovative solution to this problem, providing mathematical understanding as to why method combinations fail, or proving that some cases simply cannot be solved without further assumptions. My main comment is that while I did learn some empirical fact from the paper, I did not gain any new understanding about the problem, or learn about a novel solution to it. The initial insight is interesting and poses a good motivation for making a contribution on the topic, but I do not think that this insight alone leads to better understanding. This is my personal point of view, and it is reasonable if the authors or other reviewers hold a different one. Thanks again for the discussion.
>
> Thanks for the further discussion!  As you said, [B] highlighted why domain generalization methods find label noise challenging, but they did not try to solve the problem.  This is exactly the avenue our paper makes it's contribution, addressing both label noise and domain generalization.  So when you say you'd like to provide some understanding of why some combinations fail, this is exactly the type of thing we provide in our paper, and perhaps highlighted more clearly at the start of Section 3.3.4 and in the conclusion (with the changes marked in red).  To summarize, when trying to address our task, distinguishing between label noise and domain shifts is challenging. Thus, regularization-based techniques that largely avoid that task tend to work better as opposed to say, sample selection methods like UNICON.  The issue as highlighted in Section 3.3 is that sample selection methods tend to overfit to some domains, resulting in a biased predictor being learned.  Thus, in cases of higher domain shift (VLCS and NAG-Fashion), UNICON hurts performance.  However, this can be alleviated, as shown in Table 2, by assuming you have domain labels during training, which would reduce the effect of domain shifts.  In cases of small domain shifts (e.g., CHAMMI-CP) sample selection methods like UNICON can help performance when combined with regularization-based techniques that improve generalization (but note that these regularization-based techniques are still required).
>
> The above insights are not shown in prior work, as those papers did not attempt to solve the problem as opposed to simply highlighting that the challenge exists with respect to DG methods.  However, we not only discussed the problem, but tried to fix it with LNL methods and discovered the reason behind why some LNL methods do not combine well and a potential solution to this new problem.  These appear to follow the guidelines the reviewer put forth, and even if the reviewer now seems to argue these are obvious, the fact they did not appear in prior work suggests they are only obvious in hindsight as several other researchers discussed this topic without presenting a solution.  If we misunderstood your comment we apologize, as this response still did not provide any concrete guidance we could follow to improve our paper.  For example, a specific missing experiment or a contribution you felt was not supported enough with a specific argument as to why (e.g., by citing overlapping contributions in related work).  Without this information we would have no way of addressing this comment as general comments about novelty that are not backed up with specific examples provide little guidance for researchers or even for the area chair to make their final recommendations. Thank you again for your time to discuss this matter, we greatly appreciate it!
>
> >Thank you for the response. My small suggestion was to have the difference in the names reflect the actual difference between the methods. If the difference between the methods is their robustness to label noise, then it seems like Domain-shift methods and Domain+Label-shift methods, or anything along that line makes more sense than the current naming. However, this is not a crucial point in my view, just a small suggestion to improve the paper. Thanks again.
>
> Thank you for the suggestion, we would be happy to revise our paper to incorporate this comment.

---

### Meta-Review · Area_Chair_mXq8 · 2024-12-14

**Metareview:**

The major issue is novelty as the submission simply combined two well known research areas, learning with noisy labels and domain generalization. As the authors claimed: "Methods focused on Learning with Noisy Labels (LNL) may help with noise, but they often assume no domain shifts. In contrast, approaches for Domain Generalization (DG) could help with domain shifts, but these methods either consider label noise but prioritize out-of-domain (OOD) gains at the cost of in-domain (ID) performance, or they try to balance ID and OOD performance, but do not consider label noise at all." However, this motivation is not strong enough to justify novelty of a simple combination of learning with noisy labels and domain generalization. All the four reviewers were negative and they shared the issue of novelty I have mentioned above.

**Additional Comments On Reviewer Discussion:**

The rebuttal partially addressed the concerns from the reviewers.

---

### Decision · Program_Chairs · 2025-01-22

Reject